# Functional dynamic genetic effects on gene regulation are specific to particular cell types and environmental conditions

Anthony S Findley[1], Alan Monziani[1], Allison L Richards[1], Katherine Rhodes[2], Michelle C Ward[3†], Cynthia A Kalita[1], Adnan Alazizi[1], Ali Pazokitoroudi[4], Sriram Sankararaman[4,5,6], Xiaoquan Wen[7], David E Lanfear[8], Roger Pique-Regi[1,9]*, Yoav Gilad[2,3]*, Francesca Luca[1,9]*

[1]Center for Molecular Medicine and Genetics, Wayne State University, Detroit, United States; [2]Department of Human Genetics, University of Chicago, Chicago, United States; [3]Department of Medicine, University of Chicago, Chicago, United States; [4]Department of Computer Science, UCLA, Los Angeles, United States; [5]Department of Human Genetics, UCLA, Los Angeles, United States; [6]Department of Computational Medicine, UCLA, Los Angeles, United States; [7]Department of Biostatistics, University of Michigan, Ann Arbor, United States; [8]Center for Individualized and Genomic Medicine Research, Henry Ford Hospital, Detroit, United States; [9]Department of Obstetrics and Gynecology, Wayne State University, Detroit, United States

*For correspondence:
rpique@wayne.edu (RP-R);
gilad@uchicago.edu (YG);
fluca@wayne.edu (FL)

Present address: †Department of Biochemistry and Molecular Biology, University of Texas Medical Branch at Galveston, Galveston, United States

**Competing interest:** The authors declare that no competing interests exist.

**Abstract** Genetic effects on gene expression and splicing can be modulated by cellular and environmental factors; yet interactions between genotypes, cell type, and treatment have not been comprehensively studied together. We used an induced pluripotent stem cell system to study multiple cell types derived from the same individuals and exposed them to a large panel of treatments. Cellular responses involved different genes and pathways for gene expression and splicing and were highly variable across contexts. For thousands of genes, we identified variable allelic expression across contexts and characterized different types of gene-environment interactions, many of which are associated with complex traits. Promoter functional and evolutionary features distinguished genes with elevated allelic imbalance mean and variance. On average, half of the genes with dynamic regulatory interactions were missed by large eQTL mapping studies, indicating the importance of exploring multiple treatments to reveal previously unrecognized regulatory loci that may be important for disease.

## Introduction

Cells exist within complex environments, where levels of metabolites and signaling molecules can change rapidly. In order to thrive under such conditions, cells have evolved mechanisms to control gene expression in response to environmental perturbation. Variation in environmental exposure can explain variation in gene expression in human population samples (*Gibson, 2008*; *Idaghdour et al., 2010*; *Favé et al., 2018*; *Aguirre-Gamboa et al., 2016*; *Horst et al., 2016*; *Maghbooli et al., 2018*; *Wang et al., 2015*). For example, a highly correlated cluster of surfactant-related genes was found to be highly expressed in lung tissue from GTEx donors who died while on a ventilator (*McCall et al., 2016*). Similarly, thousands of gene expression differences were identified between sun-exposed and non-sun-exposed skin (*Kita and Fraser, 2016*). In addition to changing gene expression, environmental exposures are also able to alter splicing processes (*Pai and Luca, 2019*). This may be a direct

**eLife digest** The activity of the genes in a cell depends on the type of cell they are in, the interactions with other genes, the environment and genetics. Active genes produce a greater number of mRNA molecules, which act as messenger molecules to instruct the cell to produce proteins. The amount of mRNA molecules in cells can be measured to assess the levels of gene activity. Genes produce mRNAs through a process called transcription, and the collection of all the mRNA molecules in a cell is called the transcriptome.

Cells obtained from human samples can be grown in the lab under different conditions, and this can be used to transform them into different types of cells. These cells can then be exposed to different treatments – such as specific chemicals – to understand how the environment affects them. Cells derived from different people may respond differently to the same treatment based on their unique genetics. Exposing different types of cells from many people to different treatments can help explain how genetics, the environment and cell type affect gene activity.

Findley et al. grew three different types of cells from six different people in the lab. The cells were exposed to 28 different treatments, which reflect different environmental changes. Studying all these different factors together allowed Findley et al. to understand how genetics, cell type and environment affect the activity of over 53,000 genes. Around half of the effects due to an interaction between genetics and the environment and had not been seen in other larger studies of the transcriptome. Many of these newly observed changes are in genes that have connections to different diseases, including heart disease.

The results of Findley et al. provide evidence indicating to which extent lifestyle and the environment can interact with an individual's genetic makeup to impact gene activity and long-term health. The more researchers can understand these factors, the more useful they can be in helping to predict, detect and treat illnesses. The findings also show how genes and the environment interact, which may be relevant to understanding disease development. There is more work to be done to understand a wider range of environmental factors across more cell types. It will also be important to establish how this work on cells grown in the lab translates to human health.

effect of the environment altering normal splicing patterns, as well as the result of a compensatory effect initiated by the cell to overcome a stressful situation. Splicing is an important co-transcriptional and post-transcriptional process taking place within the nucleus, in which portions of pre-mRNAs called introns are removed, and others called exons are joined together to form a mature transcript (***Ule and Blencowe, 2019***). All forms of splicing require a variable set of proteins and small nuclear RNAs to form small-nuclear ribonucleoproteins (snRNPs), ultimately giving rise to the spliceosome (***Wahl et al., 2009***) which, together with a variable set of accessory proteins (***Han et al., 2017***; ***Gonatopoulos-Pournatzis et al., 2018***) finely tune splicing. Expression of both spliceosomal and accessory factors may be influenced by the environment, which means the environment is eventually able to alter splicing processes (***Richards et al., 2017***).

It is widely accepted that gene expression varies across individuals and that such variation is under genetic control. Expression quantitative trait loci (eQTL) mapping in samples from a variety of tissues has elucidated the tissue specificity of the genetic control of gene expression (***van der Wijst et al., 2018***; ***Dimas et al., 2009***; ***Flutre et al., 2013***). Specifically, the GTEx consortium observed a U-shaped pattern for eQTL tissue specificity, with eQTLs tending to be either highly shared amongst tissues or highly tissue-specific (***The Gtex Consortium, 2020***). These findings underscore the importance of evaluating the genetic control of gene expression across cell and tissue types.

To address the environment-specific control of gene expression, eQTL studies performed on in vitro treated cells from many donors are commonly used. The resulting genetic variants which influence gene expression response are known as response eQTL (reQTL) and represent gene-environment interactions (G×E) for molecular phenotypes. reQTL have been identified for infectious agents, drugs, and hormones, among other stimuli/perturbations (***Knowles et al., 2018***; ***Manry et al., 2017***; ***Nédélec et al., 2016***; ***Alasoo et al., 2018***; ***Kim-Hellmuth et al., 2017***; ***Quach et al., 2016***; ***Çalışkan et al., 2015***; ***Lee et al., 2014***; ***Fairfax et al., 2014***; ***Maranville et al., 2011***; ***Mangravite et al., 2013***; ***Barreiro et al., 2012***; ***Alasoo et al., 2019***; ***Huang et al., 2021***). These studies have consistently

shown that genes with G×E on gene expression are enriched for association with complex traits. However, given the large sample sizes needed to detect reQTLs, only a small number of conditions in a limited number of cell types can be tested at one time.

Allele-specific expression (ASE) is an alternative to eQTL mapping to identify the genetic control of gene expression. Rather than associating a genetic variant with gene expression, ASE uses heterozygous sites within coding regions to measure allelic imbalance in RNA-seq reads. A significant allelic imbalance within an individual implies *cis*-regulatory variation, as the *trans*-environment is constant. Similar to reQTL mapping, differences in ASE between environmental conditions is indicative of G×E, and we refer to such SNPs as conditional ASE (cASE) (*Moyerbrailean et al., 2016*). cASE has important implications for human health. For example, nearly 50% of genes with cASE were involved in complex traits by GWAS, which is significantly greater than ASE or eQTL genes (*Moyerbrailean et al., 2016*). Unlike reQTL mapping, ASE can identify G×E in small sample sizes, thereby allowing for interrogation of a broader spectrum of environmental exposures (*Moyerbrailean et al., 2016*).

While gene expression is governed by both genetic and environmental factors, it is subject to fluctuations due to stochastic factors (*Raser and O'Shea, 2004*). This noise in gene expression is gene-specific, dependent on promoter elements, and can be affected by genetic mutation (*Raser and O'Shea, 2004*; *Mogno et al., 2010*), indicating its importance in gene regulation. Indeed, there has been selection to minimize gene expression noise in yeast (*Lehner, 2008*), and gene expression variation has been linked to differential gene expression in response to perturbation in flies and humans (*Sigalova et al., 2020*). While the importance of variation in gene expression has been clearly established, much less is known on variation in the genetic control of gene expression.

Taking advantage of the reduced sample size needed to measure genetic effects by ASE, we have established a system to measure the genetic effects on gene expression in three cell types from six individuals across a large number of environmental conditions. This study design allows us to simultaneously quantify context (cell type and environment) dependent and genetic effects on gene expression. In addition to identifying thousands of instances of cASE, we have partitioned the variance in gene and allelic expression into individual, treatment, and cell type components, illustrating how each of these components can influence genetic control.

## Results

### High-throughput scan of gene expression response in 84 different contexts

We have reprogrammed lymphoblastoid cell lines (LCLs) from six individuals into induced pluripotent stem cells (IPSCs), which were further differentiated into cardiomyocytes (CMs) (*Figure 1A*). Our study was performed in batches on 96-well plates. Each batch consisted of the same cell type from three individuals, with 28 treatments and two controls. Each experiment was performed in duplicate for a total of 12 batches. (*Figure 1*). In order to shift CM cellular metabolism from fetal-associated glycolysis to adult-associated aerobic respiration, we replaced the cell culture media with galactose-containing media on day 20 (*Rana et al., 2012*; *Ward and Gilad, 2019*). We assessed the purity of CM cultures on days 25 and 27 using flow cytometry to measure the percentage of cells expressing cardiac troponin 2 (TNNT2). Purities ranged from 44.9 to 95% (LABEL:SuppTable14). CMs were derived in a single differentiation experiment per individual, so all experiments from CMs in a single individual have identical purities. Principal components analysis on gene expression reveals three distinct clusters corresponding to the three cell types (*Figure 1—figure supplement 1*). Additionally, to further verify the identity of the three cell types, we evaluated the expression of LCL, IPSC, and CM marker genes (*Figure 1—figure supplement 2*), including TNNT2, which, as expected, showed high expression in CM samples and low expression in IPSC and LCL samples (*Figure 1—figure supplement 2C*).

We exposed all cell lines to 28 different treatments (LABEL:SuppTable1), resulting in 84 cell type/treatment combinations of cellular contexts. Treatments included hormones, common drugs, vitamins, and environmental contaminants, among others. We used a two-step sequencing approach (*Moyerbrailean et al., 2015*) to identify changes in gene expression, splicing, and allelic expression across cellular contexts. First, we performed an initial shallow RNA-sequencing step (median depth of 9.5M reads) to identify treatments which caused a significant response, indicated by changes in gene expression when comparing each treatment to its vehicle control with DESeq2 (*Love et al., 2014*)

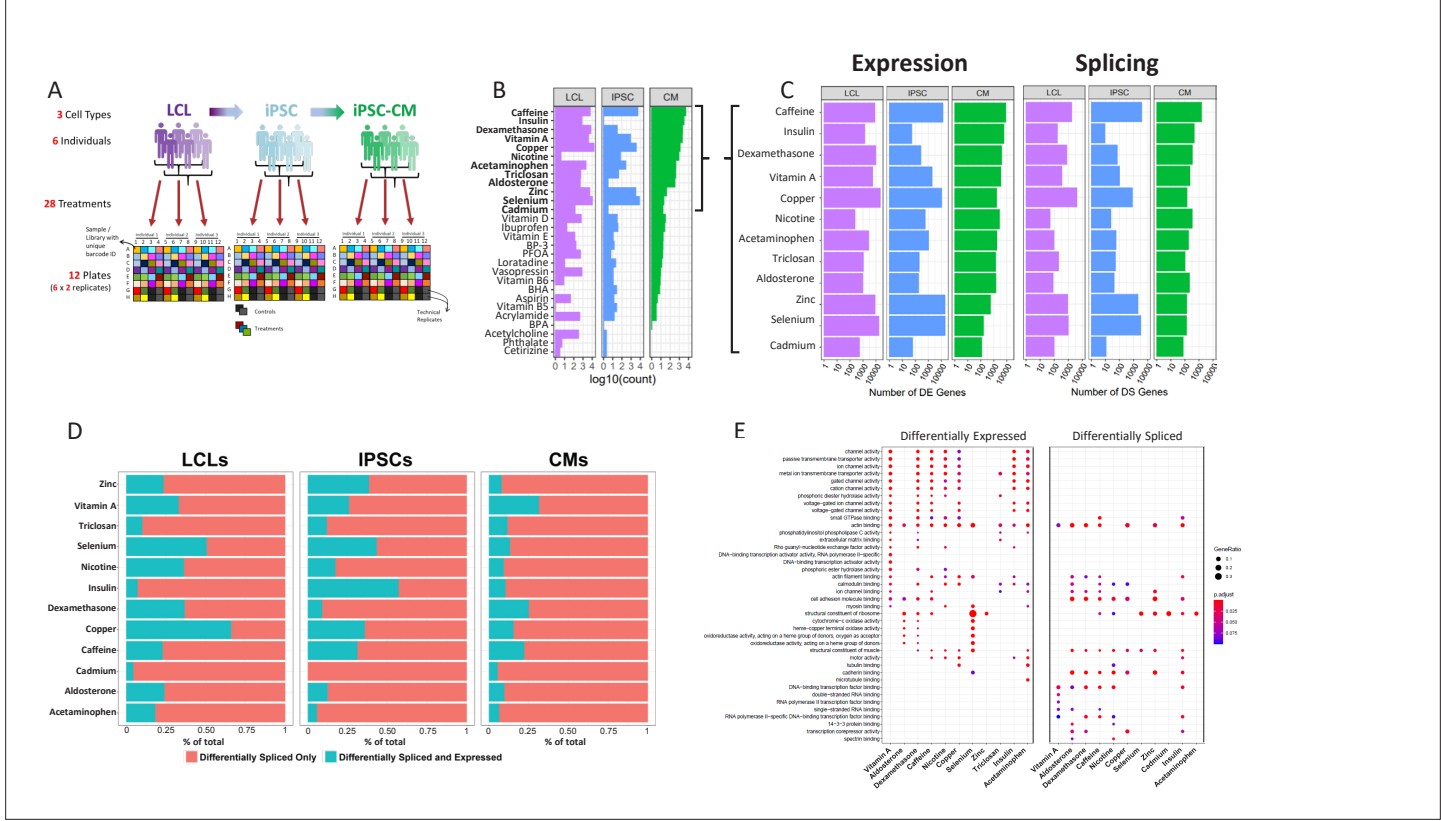

**Figure 1.** Gene expression and splicing changes in response to environmental perturbation (**A**) Study Design. We reprogramed LCLs from six donors into IPSCs, which were further differentiated into CMs. We treated all cell lines with 28 treatments and two vehicle controls, performing RNA-seq 6 hr after treatment. Three individuals per cell line were treated on one plate, and each plate was performed in duplicate, for a total of 12 plates. (**B**) Shallow Sequencing. Number of differentially expressed genes (DEGs) per treatment in each cell type from the shallow sequencing. Treatments in bold were selected for deep sequencing. (**C**) Deep Sequencing. Number of DEGs and differentially spliced genes (DSGs) per treatment in each cell type from the deep sequencing step. (**D**) Normalized stacked barplot of the number of genes that are only differentially spliced and those that are both differentially spliced and differentially expressed, for each treatment and cell type. (**E**) Dotplot generated by ClusterProfiler depicting the top enriched GO terms of CM DEGs and DSGs across environments.

The online version of this article includes the following figure supplement(s) for figure 1:

**Figure supplement 1.** Principal component analysis of gene expression.

**Figure supplement 2.** Expression of marker genes.

**Figure supplement 3.** Direction of differential splicing.

**Figure supplement 4.** Correlation between number of DEGs and DSGs.

**Figure supplement 5.** Enrichment of DSGs in DEGs.

**Figure supplement 6.** GO and KEGG enrichment dotplot for IPSCs.

**Figure supplement 7.** GO and KEGG enrichment dotplot for LCLs Dotplot generated by ClusterProfiler depicting the top enriched GO (**A**) and KEGG (**B**) terms of CM DEGs and DSGs across environments, as well as within a background list composed by all tested genes.

**Figure supplement 8.** KEGG and DGN enrichment dotplot for CMs Dotplot generated by ClusterProfiler depicting the top enriched KEGG (**A**) and DGN (**B**) terms of CM DEGs and DSGs across environments, as well as within a background list composed by all tested genes.

(*Figure 1B*, LABEL:SuppTable7). To characterize changes in gene expression, splicing, and allelic expression, we selected 12 of the 28 treatments which induced a substantial change in global gene expression for a total of 36 different contexts (in bold in *Figure 1B*, see methods).

## Gene expression and splicing response across treatments and cell types

From the 12 deep-sequenced treatments across all cell types (36 contexts and six controls, median depth/sample: 146M for LCLs, 148M for IPSCs, 273M for CMs, see LABEL:SuppTable13), we identified

differentially expressed genes between each treatment and its vehicle control in each cell type using DESeq2 (*Figure 1C*, LABEL:SuppTable8). We found between 53 (insulin in IPSCs) and 21,593 (copper in LCLs) differentially expressed genes (DEGs; FDR < 10%) out of 58,300 genes. We confirmed that the treatments were having physiological effects by examining gene ontology enrichment of differentially expressed genes. For example, as expected glucose catabolic pathways were enriched (FDR = 0.40%) in genes upregulated in response to insulin in CMs relative to all genes expressed in CMs (LABEL:-SuppTable2). Also, divalent metal ion transport (FDR = 1.8%) and transition metal ion transport (FDR = 2.8%) were enriched in genes upregulated in response to copper in LCLs. When we considered a model with all cell types together, we identified 4835 genes that show evidence of cell type × treatment interaction effects on gene expression as measured by a likelihood ratio test in DESeq2 (FDR < 10%, LABEL:SuppTable17, see Materials and methods Equation 2). These cell type × treatment interactions indicate that for these genes, the transcriptional response to the environment varies by cell type. Inter-individual variation within the regulatory sequences governing these cell-type-specific responses could lead to G×E in a cell-type-dependent manner. Interestingly, 96% of differentially expressed genes in CMs in response to treatment with insulin are not differentially expressed in LCLs or IPSCs. These unique insulin-CM genes are enriched for cholesterol and ADP metabolic processes. Together these results highlight the importance of studying environmental effects on gene expression in different cell types to avoid missing physiologically important cell type × treatment effects.

To comprehensively identify global shifts in splicing patterns across cell types, individuals, and environments we employed LeafCutter (*Li et al., 2018*), an annotation-free splicing detection tool based on intron excision from pre-mRNAs. We identified 22,334,519 unique introns, corresponding to 101,450 different transcripts and 14,066 different genes among all cell types and treatment exposures. We found between 9 (insulin in IPSCs) and 4106 (copper in LCLs) differentially spliced genes (DSGs; FDR < 10%) between treatment and control in each cell type. As one would expect, we identified treatments triggering consistent splicing alterations across all three cell types, as well as treatments predominantly having an effect in just one or two of them (*Figure 1C*). For instance, selenium exposure leads to splicing alterations in all three cell types, particularly prominent in IPSCs (3223 DSGs), while in LCLs and CMs the number of differentially spliced genes is much lower (1006 and 130 DSGs, respectively). Intron excision can be directional, meaning a given intron can be more or less excised in any given condition, with respect to baseline levels (*Richards et al., 2017*). This may result from a treatment inducing a concerted regulation of splicing towards, for example, intron retention across genes. To investigate this phenomenon, in each condition (i.e. treatment-cell type combination), we extracted the percent spliced in ($\Psi$) values and performed a two-sided binomial test to determine whether observed excision direction significantly shifts from a random 50:50 distribution. We identified 10 out of 36 environments (~29%) that show a consistent shift (FDR < 10%) in one direction of intron splicing (*Figure 1—figure supplement 3*). More specifically, four of them were identified in LCLs (insulin, selenium, zinc, copper), two in IPSCs (caffeine, zinc), and four in CMs (aldosterone, nicotine, dexamethasone, insulin). When considering the same treatment across different cell types, we found varying consistency in splicing direction. For instance, zinc treatment is associated with an increase in intron excision events in all the three cell types considered, although being significant only in LCLs and IPSCs. Response to insulin, instead, is characterized by an increase in intron retention in CMs and intron excision in LCLs. This would suggest that interaction between the environment and cellular contexts eventually determines whether cells tend to overall retain or excise introns.

## Environment triggers distinct changes in gene expression and alternative splicing

Cells are continuously exposed to different stimuli and environments, even in physiological conditions. Responses to those exposures are multi-layered, involving changes at both the transcriptional and post-transcriptional level. Because of that, differentially expressed and differentially spliced genes may be involved in distinct processes, but ultimately contributing to restoring cellular homeostasis. To understand how gene expression changes and alternative splicing contribute to the cellular response, we investigated whether the same genes are DEG and DSG in a certain context. Overall, the number of DSGs is significantly correlated with the number of DEGs (*Figure 1—figure supplement 4*, $\rho = 0.91$, $p = 1.4 \times 10^{-14}$), thus indicating that the extent of the environmental effects on the transcriptome is similar for gene expression and splicing. DSGs are also generally enriched

for DEGs, except for notable examples, such as insulin in CMs (*Figure 1—figure supplement 5*). On average 12.6% of the DSGs were found to be also DEGs in a given condition (*Figure 1D*). Overall, this suggests that there are both co-transcriptional and also independent mechanisms controlling pre and post-transcriptional responses. Furthermore, gene regulatory mechanisms involved in different biological processes responding to the environment likely modify transcription by splicing or changes in expression independently. We identified different biological processes affected by differential gene expression and splicing in response to the treatments in the three cell types (*Figure 1E*, *Figure 1—figure supplement 6* and *Figure 1—figure supplement 7*). DEGs in CMs are enriched for biological processes related to ion channel activity and transmembrane signaling across seven treatments (vitamin A, dexamethasone, caffeine, nicotine, copper, insulin, and acetaminophen), whereas DSGs were enriched for cytoskeletal protein binding across seven treatments (aldosterone, dexamethasone, caffeine, nicotine, copper, zinc, and insulin). Similarly, IPSCs showed a difference, with DEGs being involved in ion channel activity in seven treatments (vitamin A, caffeine, nicotine, copper, selenium, zinc, and acetaminophen), whereas DSGs were enriched for DNA binding and RNA biology terms across all treatments. On the other hand, both DEGs and DSGs in LCLs were highly enriched for cancer and viral-related processes, pathways and diseases across most treatments, without showing any specific difference between DEGs and DSGs. We also considered disease-gene network annotations and found that DEGs and DSGs in CMs are both enriched for genes linked to cardiovascular diseases, including different forms of cardiomyopathies and vascular disorders across all treatments except insulin for DSGs (*Figure 1—figure supplement 8*).

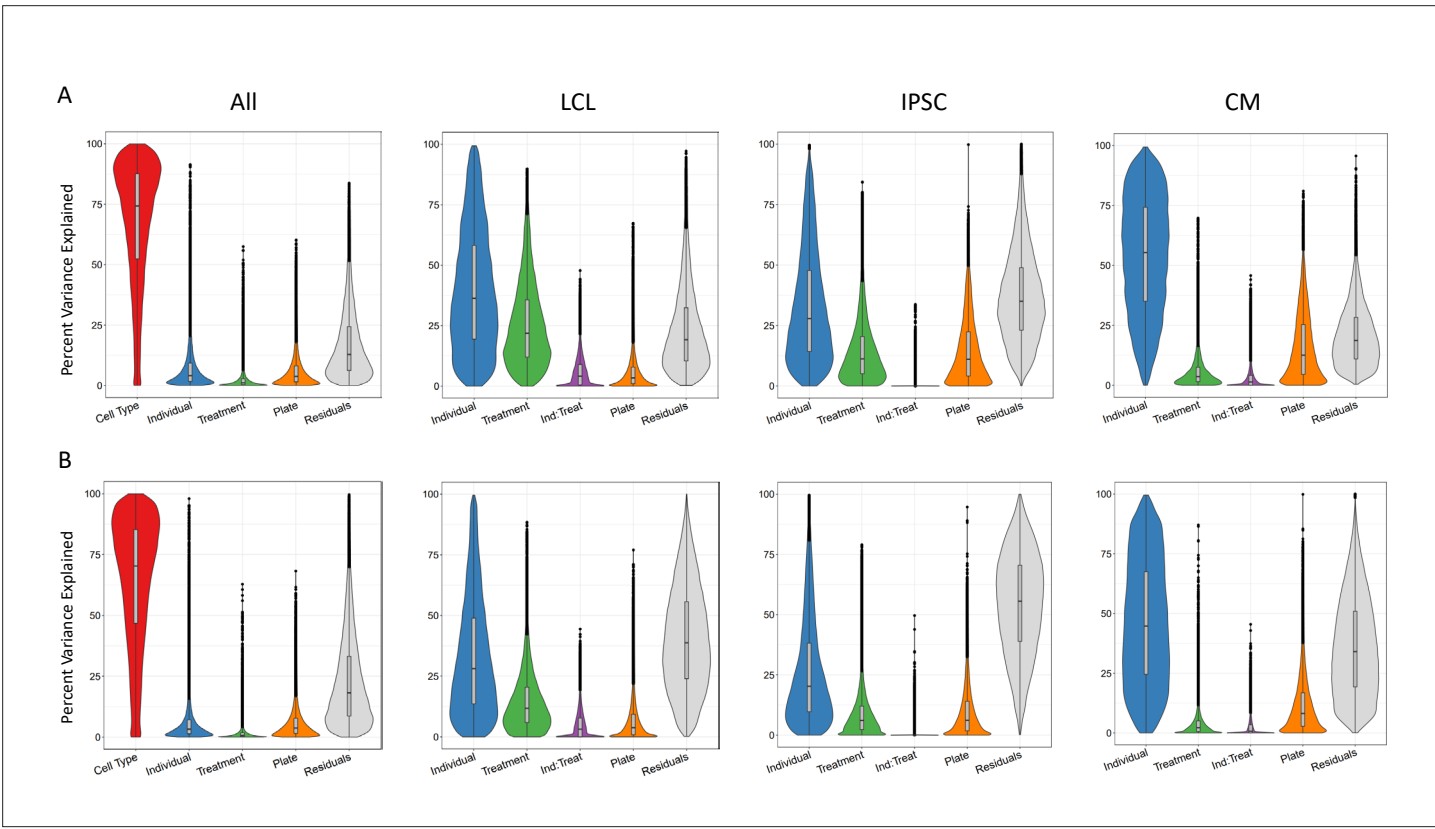

**Figure 2.** Sources of variation in gene expression and splicing. Variance for gene expression (**A**) and splicing (**B**) was partitioned into categories (horizontal axis).

The online version of this article includes the following figure supplement(s) for figure 2:

**Figure supplement 1.** Variance partitioning of gene expression and splicing in CMs with TNNT2 expression.

**Figure supplement 2.** Gene expression residual variance by presence or absence of TATA box.

## Sources of variation of gene expression and splicing

In order to identify the determinants of gene expression and splicing variation, we partitioned the variance in both gene expression and intron excision (*Hoffman and Schadt, 2016*). In particular, we were interested in uncovering the extent of the contribution of genetics, environmental and cell type effects to gene expression and intron excision. Variance which cannot be attributed to any of these defined components is counted as residual, which can be generally interpreted as due to stochasticity in the transcription process, technical variation, and/or unknown variables (e.g. cell cycle stage).

First, we considered the relative contribution of cell type, treatment, individual and batch effects (i.e. plate) to the overall variance. To this end, we partitioned the variance of all deep-sequenced samples. As expected, cell type identity has the strongest effect on both gene expression and splicing variance across samples for most genes (variance explained median value 74% and 70%, respectively, which is consistently bigger than the batch effect), with a relatively small contribution from the individual and treatment (*Figure 2*). Within each cell type, we considered treatment, individual, batch effects (i.e. plate), and we also considered the effect of any interaction between individual and treatment. These interactions should capture potential G×E effects for a specific treatment, but may also include epigenetic interactions. As expected, once we removed cell type effects, we observed a larger contribution from the other factors to both gene expression and splicing variation across samples. Similar patterns were observed for both gene expression and splicing. For example the contribution of the individual is largest in CMs (55% gene expression, 45% splicing), followed by LCLs (36% gene expression, 28% splicing) and IPSCs (28% gene expression, 20% splicing), similar to what was shown previously for gene expression (*Banovich et al., 2018*). To investigate whether this result may reflect variation in the purity of the CMs, we considered the expression of the gene *TNNT2* which encodes for the Cardiac muscle troponin T. The expression of this gene is used as a marker of differentiation of CMs and a surrogate of CM purity (*Ward and Gilad, 2019*). We repeated the analysis of variance for the CMs by including *TNNT2* expression as an additional factor. The results show that the proportion of variance explained by the individual component does not change and that the median percent variance explained by *TNNT2* expression is 6% (*Figure 2—figure supplement 1*). Treatment explained the greatest percentage of variance in LCLs (22% gene expression, 12% splicing), followed by IPSCs (11% gene expression, 6% splicing), then CMs (4% gene expression, 2% splicing). When we focused on the variance explained by measured variables (plate, treatment, individual and their interaction), we found that treatment (marginal and interaction with individual) explained at least half of the variance for 64% of the genes.

## Genetic effects on gene expression across cell types and treatments

To investigate the genetic control of gene expression across cell types and treatments, we identified SNPs exhibiting allele-specific expression (ASE). ASE occurs when there is a transcriptional imbalance between the maternal and paternal copies of an allele. Because ASE is measured within each sample, *trans* factors are kept constant, so any differences in allelic expression is most likely due to *cis*-regulatory variants. We used QuASAR (*Harvey et al., 2015*) to identify heterozygous genotypes and to provide an initial estimate of the ASE (LABEL:SuppTable3). In total, we quantified ASE at 282,278 unique SNPs in 22,397 genes. The number of SNPs with ASE across treatments within each cell type ranges from 612 to 1052 for LCLs, 607-1342 for IPSCs, and 812-1310 for CMs (*Figure 3A*).

For each individual, we measured ASE in up to 84 experimental conditions (3 cell types, 14 treatments, 2 technical replicates). This gives us an unprecedented opportunity to determine the factors that modify allelic expression in an individual. To avoid excluding conditions where an allele may be lowly expressed, we tested for ASE using a new linear model (see Materials and methods), which incorporated ASE measurements across all conditions for a SNP in a given individual, for a total of 69,205 SNPs (10,142 genes) that can be tested. By doing so, we directly infer the noise inherent in ASE measurements using the linear model (ANOVA).

We first identified 15,497 (~23%) SNPs that show evidence of ASE in any condition (ANOVA test Equations 3 and 4, FDR < 10%, LABEL:SuppTable9), corresponding to 5640 genes. Reassuringly, we observed high correlation of ASE between replicates (i.e. same individual, treatment, and cell type in different batches; Spearman's correlation median=0.66), which confirms that technical effects have a limited impact and that ASE measurements are reproducible (*Figure 3B*). When we consider control conditions on the same plate, the median correlation is identical to the one between replicates, thus

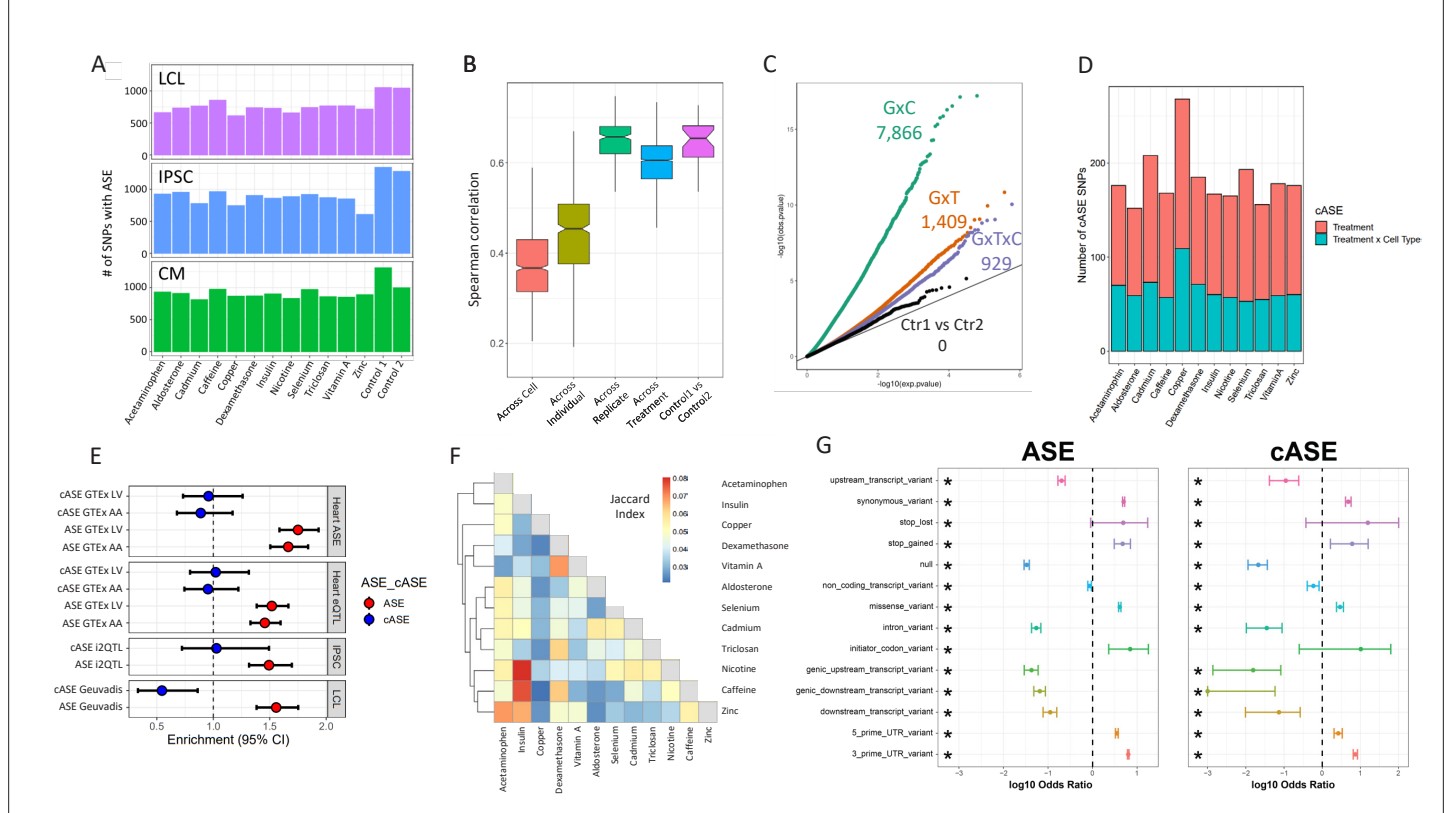

**Figure 3.** ASE and cASE (**A**) Number of SNPs with ASE in each treatment and cell type from QuASAR. (**B**) Spearman correlations between libraries across variables. The notches describe the 95% confidence interval of the median. 'Across cell' indicates comparisons between libraries which are from the same individual and treatment, but different cell types. 'Across individual' indicates comparisons between libraries from the same cell type, treatment, and replicate (i.e. plate), but different individuals. 'Across replicate' indicates comparisons between libraries from the same cell type, treatment, and individual, but different replicates (i.e. plate). 'Across treatment' indicates comparisons between libraries which are from the same cell type, individual, and replicate (i.e. plate), but different treatments. 'Control1 vs Control2' indicates comparisons between the two vehicle controls from the same individual and replicate (i.e. plate). (**C**) QQ-plot depicting interaction $p$-values for SNPs tested for cASE. G×C indicates SNPs tested for Gene×Cell type effects, G×T indicates SNPs tested for Gene × treatment effects, G×C×T indicates genes tested for Gene × cell type × treatment effects. Ctr1 vs Ctr2 indicates the coefficient in the model that accounts for the vehicle control used for each treatment. (**D**) Number of treatment G×E. Gene × treatment and gene × treatment × cell type cASE is reported for each treatment, irrespective of the cell type in which it was identified. (**E**) Enrichment of genes with ASE and treatment cASE in eGenes from large ASE/eQTL mapping studies: (1,2) CMs in GTEx heart tissues ASE/eQTL (left ventricle, LV and atrial appendage, AA), (3) IPSCs in i2QTL, (4) LCLs in Geuvadis. Odds ratios and 95% confidence intervals are plotted (Fisher's test). (**F**) Gene-by-treatment cASE shared between treatments. For each treatment pair, we calculate the proportion of shared cASE over the union of cASE identified in each of the two treatments (Jaccard Index). (**G**) Forest plot depicting the significant (FDR <10%) enrichment and depletion of ASE and cASE SNPs with respect to all SNPs tested for different genomic annotations.

The online version of this article includes the following figure supplement(s) for figure 3:

**Figure supplement 1.** Enrichment of cASE in DSGs and DEGs cASE is matched by treatment to DSGs and DEGs.

**Figure supplement 2.** cASE per SNP.

**Figure supplement 3.** Enrichment and depletion of ASE/cASE SNPs within different genomic regions.

confirming that the two control conditions can be considered technical replicates of each other. Finally, we observe a significant decrease in the correlation between ASE measurements when we consider different treatments or different cell types, which indicates G×E effects ($p < 10^{-16}$, Kolmogorov-Smirnov test).

We then used the linear model to identify significant cell-type-specific effects, treatment-specific effects, and cell type × treatment effects on ASE, which we refer to as conditional ASE (cASE)(ref. Materials and methods, Equation 5, LABEL:SuppTable10). We identified 7866 instances of cell type cASE (5452 unique SNPs, FDR < 10%), corresponding to gene × cell type interactions; 1409 instances of treatment cASE (1102 unique SNPs, FDR 10%), corresponding to gene × treatment interactions;

and 929 instances of cell type × treatment cASE (715 unique SNPs, FDR < 10%), corresponding to gene × cell type × treatment interactions (*Figure 3C and D*). When we considered genes with at least one SNP with ASE, we found 2822 unique genes with gene × cell type interactions, 979 unique genes with gene × treatment interactions, and 689 unique genes with gene × cell type × treatment interactions.

We next investigated whether these genetic effects on gene expression have been previously observed in large scale eQTL mapping studies that largely ignored dynamic regulatory interactions. For this analysis, we considered the CMs and their tissue counterparts in GTEx (left ventricle and atrial appendage), the LCLs and the GEUVADIS dataset (*Lappalainen, 2013*), and IPSCs and the I2QTL consortium data (*Bonder et al., 2021*). In CMs, we identified 3033 genes with ASE. Of these genes, 50% (1519) and 52% (1619) were eGenes in left ventricle and atrial appendage from GTEx, respectively. This translates to a 1.52- and 1.46-fold enrichment of ASE genes in GTEx genes in the left ventricle and atrial appendage, respectively ($p < 10^{-16}$, Fisher's exact test, *Figure 3E*). Of the 338 genes with treatment cASE in CMs, 170 and 178 are eGenes in left ventricle and atrial appendage in GTEx. Interestingly, treatment cASE genes in CMs were not significantly enriched in either tissue (odds ratio = 1.02, 0.95). In addition to eQTL mapping, GTEx also used ASE to measure *cis*-regulatory effects (*Castel et al., 2020*). As with GTEx eGenes, GTEx left ventricle and atrial appendage genes with ASE were enriched for CM ASE genes (odds ratio = 1.75, 1.66 for each tissue, respectively, $p < 10^{-16}$), but not CM cASE genes. A similar pattern was seen in LCLs and IPSCs. In LCLs, there were 3237 genes with ASE, and 24% (764) were eGenes in Geuvadis (1.55-fold enrichment, $p = 1.4 \times 10^{-13}$). Of the 167 genes with cASE in LCLs, 26 were eGenes in Geuvadis. This represented a significant depletion of cASE genes in Geuvadis eGenes (odds ratio = 0.55, $p = 7.7 \times 10^{-3}$). In IPSCs, there were 3113 genes with ASE, and 80% were eGenes in i2QTL (1.49-fold enrichment, $p = 1.0 \times 10^{-10}$). Of the 352 genes with cASE in IPSCs, 284 were eGenes in i2QTL. As with the CMs, these cASE genes were not significantly enriched (odds ratio = 1.03). These results indicate that investigating the control of gene expression across different environmental contexts, even in a small number of individuals, can identify new instances of genetic regulation that are missed in large eQTL mapping studies that do not explicitly sample different environmental contexts.

To investigate whether our G×E results replicate in other environments previously investigated, we calculated the overlap between genes with gene × treatment interactions in our dataset and genes with G×E identified in fourteen previous ASE and eQTL mapping studies spanning a range of cell types and treatments. Of 979 genes with treatment cASE, 850 (87%) replicated in at least one of these datasets (with $p < 0.05$ in the original study, LABEL:SuppTable4). *Knowles et al., 2018* exposed IPSC-derived CMs to doxorubicin, a chemotherapeutic agent, and performed response eQTL (reQTL) and response splicing QTL (rsQTL) mapping. When considering genes with a reQTL, 105 could be tested for cASE in CMs and 79 of these genes had nominally significant cASE (p < 0.05), while 12 displayed cASE after multiple test correction (FDR < 10%) spanning nine different treatments in our study (LABEL:SuppTable5). For example, for three of those genes (*FRAS1*, *PDGFC*, and *MPHOSPH6*) we can now annotate a G×E interaction with caffeine in addition to doxorubicin. Out of 740 genes with one or more sQTLs, 182 and 88 have ASE and cASE in our datasets (p < 0.05), respectively, whereas of the 62 genes with response sQTLs to doxorubicin 16 have ASE and 11 have cASE (p < 0.05). Overall, these results point to different regulatory mechanisms that can lead to G×E and that are context-specific.

The number of genes with gene × treatment interactions varies with a low of 96 (93 SNPs) for aldosterone and a high of 176 (159 SNPs) for copper (1.8-fold range between minimum and maximum number of cASE). This is in contrast with the spread in gene expression and splicing responses across treatments (two orders of magnitude lower than the range between minimum and maximum number of DEGs). When we considered cASE in each cell type separately, indeed we did not find a significant enrichment or depletion for cASE genes being differentially expressed or differentially spliced between treatment and control (*Figure 3—figure supplement 1*). However, despite the number of cASE being similar across treatments, the genes with G×E are largely different. Most treatment cASE SNPs were specific to one treatment (80.5%, *Figure 3—figure supplement 2*), so there was little sharing between treatments (*Figure 3*). While this could be due to a lack of power in detecting cASE with small effect sizes, it also suggests that genes with the strongest cASE are different across conditions.

## Biological and clinical significance of allele-specific expression across cell types and treatments

SNPs with ASE and cASE may have a direct effect on gene expression or more often may indirectly reflect the effect of a regulatory variant located in non-coding regions. Regulatory sites that could be located within the gene transcript itself include those in the 3' and 5' UTRs, in splicing junctions, premature stop codons, and possibly exonic enhancers. Thus, SNPs with ASE and cASE located within these regulatory sites could be putative causal sites for the observed allelic imbalance (*Mohammadi et al., 2019*).

We retrieved the genomic location of the SNPs associated with ASE and cASE, as well as their predicted functional impact from dbSNP (https://www.ncbi.nlm.nih.gov/snp, *Sherry et al., 2001*) (*Figure 3—figure supplement 3*). When we considered all variants that could be tested for ASE, both ASE and cASE SNPs were found to be significantly enriched in 5' and 3' UTRs, and among variants classified as missense or altering a stop codon (FDR < 10%, Fisher's exact test; *Figure 3G*). On the contrary, SNPs located within upstream/downstream transcript and genic regions, within introns and in splicing sites were found to be significantly depleted in ASE/cASE SNPs. Taken together, these data suggest that functional SNPs with ASE or cASE most likely exert their function by changing transcriptional and/or RNA stability regulatory sequences because the 5' and 3' regions tend to affect these processes, rather than affecting splicing processes.

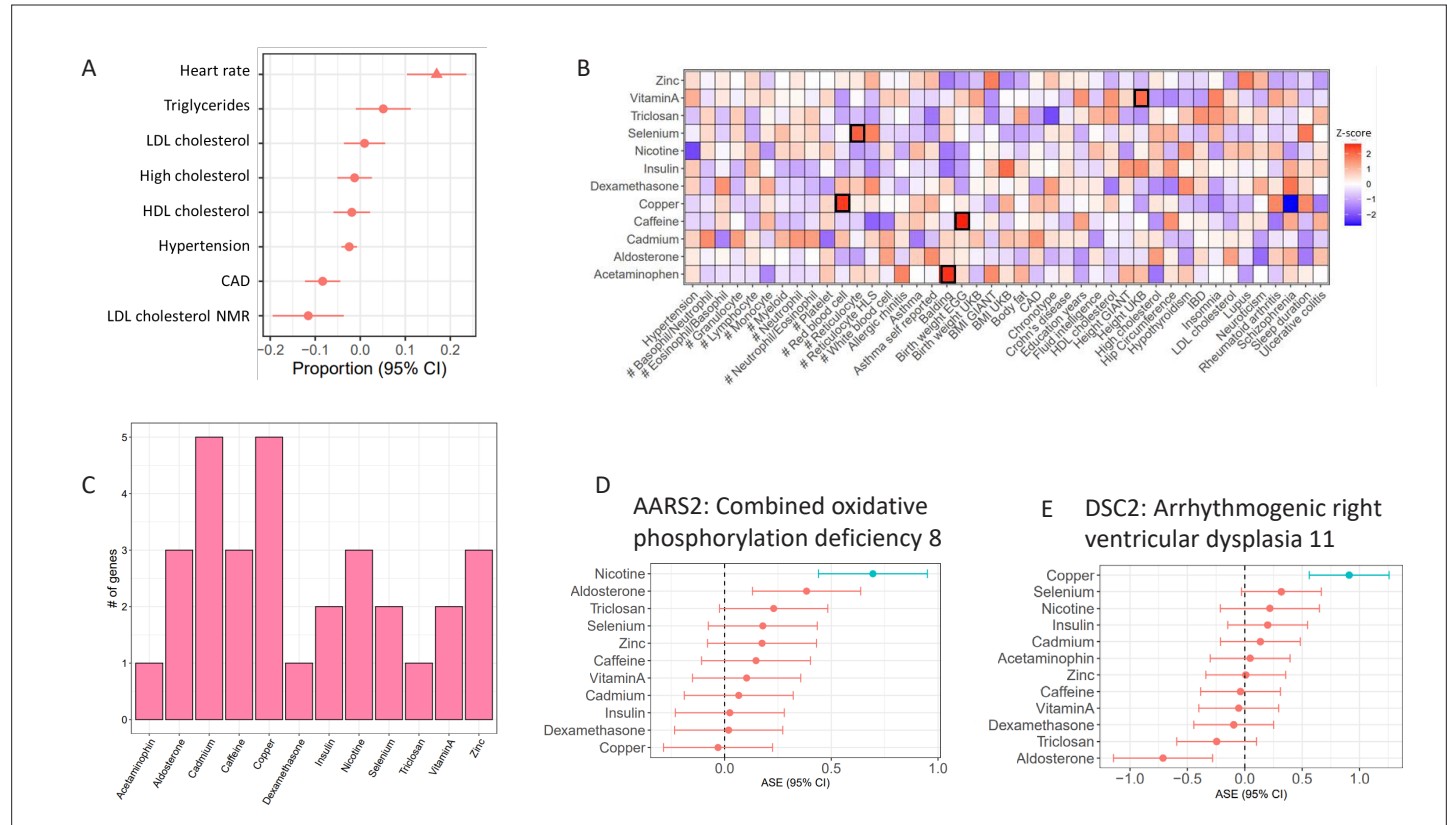

**Figure 4.** Treatment cASE genes in TWAS (**A**) CM cASE overlap for genes associated with complex traits in TWAS. For each trait, the forest plot shows the proportion of cASE genes overlapping TWAS genes, relative to the average proportion calculated across all traits. Overlap with genes associated with heart rate was significantly greater than other traits (p=0.01, FDR=0.08). (**B**) Overlap of gene-by-treatment cASE from all cell types with genes associated with complex traits in TWAS. The color of the box represents the Z-score of the one sample proportion test comparing the proportion of cASE-TWAS overlap for a treatment to the average cASE-TWAS overlap across treatments. (**C**) Gene-by-treatment cASE overlap with CARDIoGRAM TWAS. (**D,E**) Forest plots depicting change in allelic imbalance in CMs due to treatments for AARS2 and DSC2, two genes implicated in Medelian disease. Lines in blue indicate significant treatment cASE.

The online version of this article includes the following figure supplement(s) for figure 4:

**Figure supplement 1.** Overlap between atrial fibrillation GWAS prioritized genes and treatment cASE.

**Figure supplement 2.** Enrichment of treatment cASE annotations in 22 GWAS.

To investigate the role of genes with cASE in human phenotypes, we considered genes associated with complex traits. To this end, we used the results from Probabilistic Transcriptome Wide Association Studies (PTWAS) (*Zhang et al., 2020*), which combined eQTL data from GTEx and GWAS data from several large scale studies to identify which genes are most likely to be in the causal pathway for complex traits. First, we considered eight cardiovascular traits, including blood lipids, hypertension, coronary artery disease, and heart rate. Heart rate was the only trait with a significant overlap with CM-specific cASE, compared to the average overlap with other cardiovascular traits (FDR < 10%). Interestingly, heart rate is the trait among those tested where cardiomyocytes have a direct physiological role (*Figure 4A*). We then investigated the role of G×E in a larger panel of 45 complex traits (*Figure 4B*). Five traits were enriched for treatment cASE genes in at least one condition (p < 0.05) compared to the other treatments. For example, genes associated with low birthweight are enriched for genes that have caffeine cASE (*Figure 4B*). For the 428 genes associated with coronary artery disease (CAD), cASE for metal treatments have the greatest overlap, with cadmium and copper each having five cASE genes associated with CAD, and zinc having three (*Figure 4C*). Interestingly, cadmium causes endothelial dysfunction and promotes atherosclerosis (*Messner et al., 2009*), while copper deficiency has been linked to the development of CAD (*DiNicolantonio et al., 2018*). *FAM213A* (also known as *PRXL2A*), which is involved in redox regulation (*Guo et al., 2015*; *Xu et al., 2010*) is shared as treatment cASE for cadmium and copper, in addition to being treatment cASE for nicotine.

We also considered an atrial fibrillation GWAS which identified 151 genes near associated loci (*Nielsen et al., 2018*). Ten of these genes displayed treatment cASE in at least one condition, for a total of 21 gene-cASE condition pairs (*Figure 4—figure supplement 1*, LABEL:SuppTable12). The greatest overlap was with dexamethasone cASE. Corticosteroids are antiinflammatory drugs which share molecular pathways with inflammatory disease. Inflammation is a risk factor for atrial fibrillation (*Hu et al., 2015*; *Harada et al., 2015*). Additionally, in multiple population-based, case-control studies, corticosteroid use was associated with new-onset atrial fibrillation (*Van Der Hooft et al., 2006*; *Christiansen et al., 2009*). Unlike atrial fibrillation, GWAS for heart failure have not identified many association signals, likely due to the highly heterogeneous nature of the disease. Two recent meta-analyses identified 3 and 13 genes, respectively (*Arvanitis et al., 2020*; *Shah et al., 2020*). Three of the genes associated with heart failure had treatment cASE: *FAM241A* in five conditions (copper, dexamethasone, insulin, caffeine, and vitamin A); *BAG3* in five conditions (dexamethasone, caffeine, vitamin A, nicotine, and aldosterone); and *KLHL3* in triclosan.

While most causes of heart disease are not due to a mutation in a single gene, regulatory variation controlling the expression of Mendelian disease genes can affect complex trait risk (*Freund et al., 2018*). In CMs, we found three genes in OMIM (Online Mendelian Inheritance in Man) (*Hamosh et al., 2005*) with gene × treatment interactions that are known to cause Mendelian forms of heart disease: *PSMA6*, *AARS2*, *DSC2*. *AARS2* displayed cASE in response to nicotine and mutations in this gene cause Combined Oxidative Phosphorylation Deficiency 8, which manifests as fatal infantile hypertrophic mitochondrial cardiomyopathy (*Figure 4D*). *DSC2*, instead, displayed cASE in response to copper and mutations in this gene cause Arrhythmogenic right ventricular dysplasia 11 (*Figure 4E*).

Much of complex trait heritability is not explained by genome-wide significant variants, but rather results from the contribution of many variants with smaller effect sizes (*Manolio et al., 2009*; *Yang et al., 2010*; *Boyle et al., 2017*). Methods to partition the heritability of complex traits can be used to determine the proportion of heritability attributable to genomic annotations (*Yang et al., 2011*; *Finucane et al., 2015*). We used RHE-mc (*Pazokitoroudi et al., 2020*) to partition the heritability of complex traits from UK Biobank based on cASE annotations. First, we partitioned the heritability of 17 traits, using an annotation of genes with cell type cASE for CMs. CM cell type cASE was significantly enriched for diastolic blood pressure, autoimmune disease, and height. We then considered 22 traits, using treatment cASE as an annotation (*Figure 4—figure supplement 2*). Diastolic blood pressure was enriched for vitamin A, systolic blood pressure was enriched for vitamin A, copper, and zinc. Respiratory disease was enriched for caffeine, and smoking status for selenium.

## Characterizing variation in genetic regulation of gene expression

The presence of cASE indicates that the genetic control of gene expression varies significantly with changes in the environment. To further investigate the role of the environment and cell type

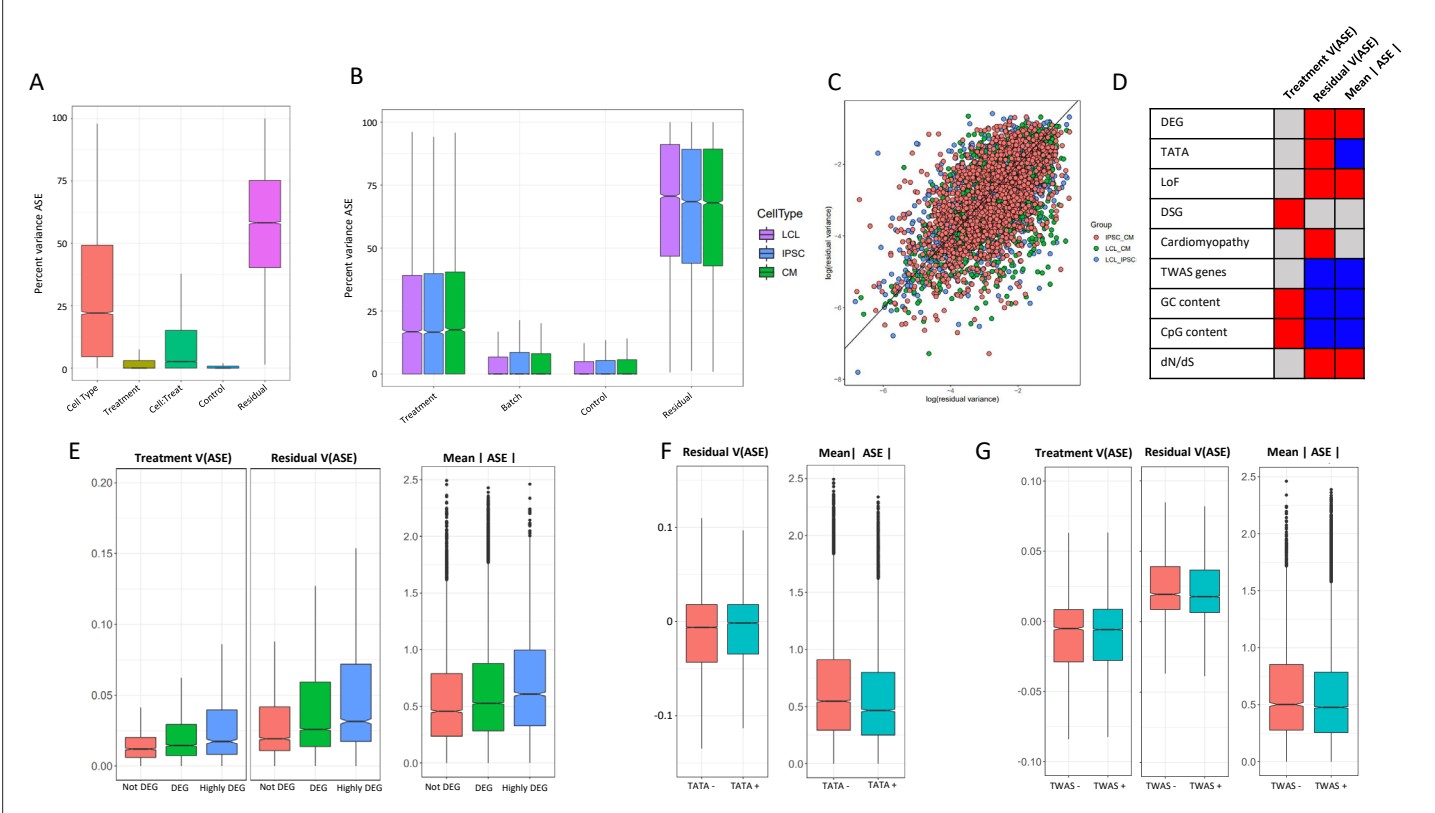

**Figure 5.** ASE residual variance (**A**) Variance partitioning of ASE across all cell types. (**B**) Variance partitioning of ASE within each cell type separately. (**C**) ASE residual variance across cell types. For SNPs with ASE variance measured in more than one cell type in the cell-type-specific model, we have plotted the residual variance in each pair of cell types. (**D**) Relationships between eight gene annotations and treatment ASE variance, residual ASE variance, and mean absolute ASE. Red indicates that the annotation is positively correlated to ASE variance or absolute ASE, while blue indicates a negative correlation. Gray indicates no significant relationship. (**E**) Treatment variance, residual variance, and mean absolute ASE by DEG. We have separated genes into three categories: (1) Not DEG: Not differentially expressed in response to any treatment. (2) DEG: Differentially expressed with log$_2$ fold change > 0.25 in at least one treatment but not greater than 0.75 in any treatment. (3) Highly DEG: Differentially expressed with log$_2$ fold change > 0.75 in at least one treatment. Each gene can only be in one category. (**F**) Residual ASE variance and mean absolute ASE for genes with or without a TATA box in the promoter. (**G**) Treatment variance, residual variance, and mean absolute ASE for genes significant or not significant by TWAS.

The online version of this article includes the following figure supplement(s) for figure 5:

**Figure supplement 1.** Gene ontology for ASE variance.

**Figure supplement 2.** Treatment variance of DSGs.

in controlling ASE, we used a mixed-effects model to partition the variance in ASE for each SNP-individual pair across all experimental conditions, similar to our analysis of gene expression and splicing variance (see Materials and methods). When considering all cell types together, the greatest amount of explainable variance in ASE was captured by the cell type (***Figure 5A***, LABEL:SuppTableMixed_All), similar to the results explaining variability in gene expression (***Figure 3A***). However, when considering variance in ASE within each cell type, a much larger proportion of the explainable variance was attributed to the effect of a treatment (***Figure 5B***, LABEL:SuppTableMixed_Sep). Specifically 17.5%, 16.6%, and 16.8% of variance (median) in ASE was attributed to treatment effects in CMs, IPSCs, and LCLs, respectively, compared to 3.6%, 11.2%, and 21.9% variance (median) in total gene expression.

There has been extensive investigation into gene expression stochasticity (***Sigalova et al., 2020***). However, the role of stochasticity in the genetic control of gene expression has proven more difficult to study (***Sarkar et al., 2019***). Because our experimental design measures ASE across many conditions, we can analyze the residual variance as a measure of genetic control stochasticity. We found that residual variance at a particular SNP is correlated ($r \in [0.55 - 0.59]$, $p < 10^{-16}$, 95% confidence intervals = [0.55, 0.63] for LCL vs IPSC, [0.50, 0.59] for LCL vs CM, and [0.55, 0.61] for IPSC vs CM) across cell types, indicating that unknown factors that contribute to the genetic control of gene expression

are conserved across tissues (*Figure 5C*). In LCLs, genes with high residual variance were enriched in 22 GO terms, with the greatest enrichments being related to tissue morphogenesis and developmental processes, while genes with low residual variance were enriched in 82 GO terms, including RNA processing and immune activation. In IPSCs, low residual variance genes were enriched in 128 GO terms, including peptide metabolic pathways, which was also seen in CMs. We found that low residual variance genes in CMs were enriched in 123 GO terms, including processes related to cell signalling (LABEL:SuppTable6, *Figure 5—figure supplement 1*).

To investigate the biological significance of ASE changes in magnitude and variance, we considered nine genomic features: genes differentially expressed in response to any treatment, genes differentially spliced in response to a treatment, genes with a TATA-box promoter, CpG percentage in the promoter, GC percentage in the promoter, genes which cause mendelian cardiomyopathies, genes associated with complex traits by TWAS, gene tolerance to loss of function mutations, and dN/dS ratio (the ratio of nonsynonymous to synonymous mutations in a gene, which is a measure of selective pressure) (*Figure 5D*). We first considered the distribution of allelic expression for each of these categories. DEGs were characterized by high ASE compared to genes that do not respond to the treatments (*Figure 5E*). Allelic expression was also elevated for genes tolerant to loss of function mutations, while genes associated with complex traits tend to have lower allelic expression (*Figure 5F and G*), which reflects the difference in phenotypic relevance for these gene categories. Genes with lower ratio of nonsynonymous to synonymous mutations (dN/dS, i.e. genes under negative selection) are associated with lower mean ASE. When we considered regulatory features of the genes with ASE, we found lower ASE for genes with a TATA box, and for genes with higher promoter CpG and GC content.

When focusing on variance in allelic expression, we found differential gene expression to be associated with increased residual variance, suggesting that for genes with more dramatic responses to the environment, gene expression is under a less stringent genetic control (*Figure 5E*). Differentially spliced genes, however, are associated with increased treatment variance, and not increased residual variance (*Figure 5—figure supplement 2*). We then considered promoter architecture. SNPs in genes with a TATA box had increased residual variance compared to SNPs in genes without a TATA box (*Figure 5*). We found no significant relationship between treatment variance and whether the gene contains a TATA box. In addition, GC content and CpG islands in a gene's promoter was associated with increased treatment variance and decreased residual variance. Genes which are known to cause cardiomyopathies when mutated had a small increase in residual variance ($p = 0.0032$), while genes implicated in any of the 103 TWAS in *Zhang et al., 2020* had decreased residual variance. Finally, we considered if intolerance to loss of function and dN/dS ratio could be related to ASE variance. As expected, genes that are less tolerant to loss-of-function mutations and genes under negative selection (i.e. low dN/dS) have significantly lower residual variance.

## Discussion

A fundamental open question in the field of genomics is understanding the sources of variation in gene expression across different individuals, cell types, and environmental contexts. Each of these components has been investigated separately in previous studies (*Gibson, 2008*; *Idaghdour et al., 2010*; *Favé et al., 2018*; *Aguirre-Gamboa et al., 2016*; *Horst et al., 2016*; *Maghbooli et al., 2018*; *Wang et al., 2015*; *van der Wijst et al., 2018*; *Dimas et al., 2009*; *Flutre et al., 2013*; *Knowles et al., 2018*; *Manry et al., 2017*; *Nédélec et al., 2016*; *Alasoo et al., 2018*; *Kim-Hellmuth et al., 2017*; *Quach et al., 2016*; *Çalışkan et al., 2015*; *Lee et al., 2014*; *Fairfax et al., 2014*; *Maranville et al., 2011*; *Mangravite et al., 2013*; *Barreiro et al., 2012*; *Alasoo et al., 2019*; *Huang et al., 2021*; *Moyerbrailean et al., 2016*; *Findley et al., 2019*). Here, in order to investigate the transcriptome along the three different axes, we used a study design that combines IPSC technology, high-throughput screening and allele-specific expression analyses. The cell type context has the overall strongest effect on gene expression, splicing, and allelic expression. This is observed both in terms of differences of the mean gene expression levels across conditions using fixed effect models, or in analyzing the variance on gene expression using random effect models. Importantly, we demonstrate that a large number of context-specific genetic effects on gene expression are not captured by existing large cohort eQTL studies (e.g. GTEx, GEUVADIS, i2QTL), but can be discovered even with limited sample sizes, when using an allele-specific expression study design. Our unique approach,

accounting for cell-type but also genetic and environmental influences has also revealed that environmental impact on gene expression is substantial and particularly important for genes that influence complex traits.

IPSCs are a valuable model system for studying primary cell types which are difficult to obtain and/or culture (*Sterneckert et al., 2014*). Extensive work by our group and others have shown that despite the potential introduction of reprogramming effects, IPSC-derived CMs mimic primary CMs (*Pavlovic et al., 2018*), recapitulate in vivo CM phenotypes (*Matsa et al., 2016*; *Burridge et al., 2016*; *Carvajal-Vergara et al., 2010*), retain donor-specific expression signatures (*Panopoulos et al., 2017*; *Burrows et al., 2016*; *DeBoever et al., 2017*; *Rouhani et al., 2014*), and can be used to assess the effects of environmental exposures on CM gene expression and cellular physiology (*Ward and Gilad, 2019*; *Knowles et al., 2018*; *Sharma et al., 2017*; *Kitani et al., 2019*; *Matsa et al., 2016*; *Sharma et al., 2014*; *Burridge et al., 2016*). This ability to study CMs themselves is especially important given that GWAS studies for heart failure have revealed little of the genetic architecture of the disease. While pedigree analysis suggests that the heritability of heart failure is 26 - 34% (*Lindgren et al., 2018*), two recent, large-scale GWAS identified only eleven (*Shah et al., 2020*) and three (*Arvanitis et al., 2020*) genomic loci associated with heart failure. This could be due to a lack of quantitative measurements of heart failure, complex etiologies, or the significant environmental contributions to developing heart failure (*Rau et al., 2015*). Our approach allows for interrogation of the effects of environmental exposures in a controlled environment on CM gene expression.

Some treatments had very dramatic effects on gene expression, such as copper and selenium in LCLs with more than 10,000 DEGs each. However, these large changes in gene expression were seen in our previous work (*Moyerbrailean et al., 2016*) (7869 DEGs in copper, 14,057 DEGs in selenium), and the increased sequencing depth and greater number of individuals in this study increased our power to detect DEGs. On average, we show the number of DSGs is directly correlated with the number of DEGs, therefore suggesting that environmental perturbations have a similar degree of effect on both transcriptional and post-transcriptional processes. However, only a minor fraction of genes were found to be both differentially expressed and spliced, which suggests that changes in gene expression and splicing represent distinct regulatory mechanisms by which cells can respond to their environment. Indeed, this is supported by GO and KEGG enrichment analyses, showing DSGs in CMs are involved in cytoskeletal activities, whereas DEGs were found to play a role in ion channel processes.

Partitioning the variance in gene expression and intron excision across all cell types gives a broad overview of the relative contribution of cell type and treatment components to transcription and splicing processes. There were similar patterns for gene expression and splicing. Similar to previous work on partitioning the variance in gene expression across a panel of LCLs, IPSCs, and CMs by *Banovich et al., 2018*, variance due to the individual component was greatest in CMs, followed by LCLs and IPSCs. Yet herein, we were additionally able to consider the treatment component, which was greatest in LCLs and least in CMs. One possible explanation is that the chosen treatments may have more effect in LCLs; alternatively, this may be a consequence of greater inter-individual heterogeneity in CMs compared to LCLs. Regardless of cell type, residual variance was greater for splicing than gene expression. This is consistent with previous work showing that splicing is an error-prone process with a high degree of noise (*Melamud and Moult, 2009*; *Pickrell et al., 2010*; *Wan and Larson, 2018*).

A key new feature of our study design is that it allowed us to analyze allelic expression across 12 environments (and two controls), three cell types, six individuals and two technical replicates each, for a total of 504 experimental samples. This is the largest single study of allelic expression comparing in parallel cell type effects, treatment conditions, and their interactions while controlling for technical variation. We used a new linear model to precisely estimate allelic expression and its variance directly from all the measurements of ASE for each individual (up to 84). As a result, we have a more complete view of the pervasiveness of G×E and we were able to directly investigate whether environmental effects on genetic regulation of gene expression differ across cell types. Our study design allowed us to specifically investigate interactions between genotype, cell type, and treatment. Interestingly, many treatment effects on allelic expression vary across cell types, therefore it is important to consider jointly cellular and environmental contexts. As the IPSC technology allows investigators to study multiple cell types from the same individuals, future study designs should consider the importance

of studying environmental effects across cell types to learn about pleiotropic effects of potential biomedical relevance.

Both eQTL and ASE studies have demonstrated that genetic effects on gene expression vary significantly across cellular (*The Gtex Consortium, 2020*) and environmental contexts (*Moyerbrailean et al., 2016*; *Knowles et al., 2018*; *Manry et al., 2017*; *Nédélec et al., 2016*; *Alasoo et al., 2018*; *Kim-Hellmuth et al., 2017*; *Quach et al., 2016*; *Çalışkan et al., 2015*; *Lee et al., 2014*; *Fairfax et al., 2014*; *Maranville et al., 2011*; *Mangravite et al., 2013*; *Barreiro et al., 2012*; *Alasoo et al., 2019*; *Huang et al., 2021*). However, we do not know how stable the genetic control of gene expression is across technical and environmental contexts. This cannot be investigated with eQTL mapping because the genetic effects on gene expression are estimated across individuals, but can be explored with multiple measurements of allelic expression from the same individual. We observed that residual variance of allelic expression is conserved across cell types, which indicates that the intrinsic properties controlling variation in allelic expression are consistent in different tissues and have biological significance. For example, genes which had a strong expression response to at least one treatment had increased residual variance over genes which had no response or a smaller response, while genes with a splicing response had greater treatment variance only. We hypothesize that greater residual variance indicates a less stringent control of allelic expression. This may depend on the regulatory architecture of these genes and may similarly enable large fluctuations in gene expression across contexts. Prior work on gene expression variance has demonstrated that genes with a TATA box have increased noise (*Mogno et al., 2010*). This was also shown by *Sigalova et al., 2020* as well as in our data here (*Figure 2—figure supplement 2*). However, this finding has not been evaluated for the genetic control of allelic expression. Our results indicate that genes with ASE which are under the control of a TATA box promoter have increased residual variance over ASE genes without a TATA box, and thus greater noise. TATA box promoters, as opposed to CpG island-associated promoters, have been associated with tissue-specific genes and high conservation across species (*Carninci et al., 2006*). Genes which are more tolerant to loss of function mutations showed greater allelic expression residual variance, indicating that redundancy in gene function allows for less stringent genetic control. This could allow for the evolution of new regulatory elements, resulting in new patterns of gene expression. Conversely, genes which are under negative selection (i.e. low dN/dS) have low residual variance, underscoring the importance of preserving stable expression of these genes. Tolerance to ASE variation could result in robustness against regulatory decoherence (*Lea et al., 2019*), which could be further explored in future studies. Cell type and treatment-specific effects are the largest identifiable sources of variation in allelic expression for hundreds of genes, as we demonstrated using both variance decomposition and fixed effects as statistical models.

Splicing sites have very conserved sequences, which are recognized by spliceosomal and accessory proteins which ultimately determine splicing patterns. Moreover, splicing is finely tuned by regulatory *cis* RNA sequences within both exons and introns, which are recognized by several RNA-binding proteins (RBPs). Mutations within these consensus sequences therefore have catastrophic consequences in pre-mRNA splicing, eventually being associated with a plethora of different pathological conditions (*Faustino and Cooper, 2003*; *Lukong et al., 2008*; *Anna and Monika, 2018*), including several forms of cystic fibrosis (*Friedman et al., 1999*; *Bobadilla et al., 2002*) and in both Becker and Duchenne muscular dystrophies (*Habara et al., 2009*). In addition to deleterious mutations, naturally occurring polymorphisms such as SNPs may contribute to alter the strength of splicing signals, eventually changing splicing outcomes. However, SNPs linked either to ASE or cASE were found to be mainly depleted in both donor and acceptor splicing sites, eventually suggesting that differential intron splicing is not a major mechanism underlying ASE/cASE. In fact, those SNPs were found to be enriched within several genic regions, in particular within UTRs and, interestingly, causing either the loss or gain of stop codons. These events have dramatic effects on RNA stability, as improper stop codon localization can activate the nonsense mRNA decay (NMD) pathway, resulting in transcript degradation (*Brogna and Wen, 2009*). Moreover, synonymous polymorphisms may alter the codon optimization of a given mRNA, a process which has been linked to reduced half-life (*Presnyak et al., 2015*). All these events may eventually lead to reduced abundance of the transcript carrying a particular allele, compared to the other copy, thus resulting in ASE.

We identified 3198 genes with cASE, including 2822 genes with cell type cASE, 979 genes with treatment cASE and 689 genes with treatment × cell type cASE. Many genes with treatment cASE

overlap with genes that have context-specific genetic effects discovered by other studies; yet there is little sharing for any specific pair of conditions. In our study, since we compare ASE within the same genetic background, this limited sharing of ASE between treatments strongly suggests the existence of independent regulatory variants in context specific cis-regulatory modules. eQTL mapping consortia, like GTEx, identify eQTLs under one arbitrary condition, so genetic effects on gene expression which occur only transiently during development or under certain environmental conditions will be missed (*Moyerbrailean et al., 2016*; *Strober et al., 2019*; *Findley et al., 2019*; *Resztak et al., 2021*; *Umans et al., 2021*; *Cuomo et al., 2020*). A common approach to increasing identification of eQTLs is to increase the sample size. However, interrogating additional environmental conditions will be required to gain a more complete understanding of the genetic control of gene expression. This is reflected in the lack of enrichment for cASE genes in eQTLs from large studies in three tissues/cell-types. Even with a sample size of only six individuals, we identified many new genes which are genetically regulated under specific environmental conditions. For example, in CMs approximately half (~1500) of the ASE genes were not eGenes in GTEx heart tissues. They were identified as ASE genes in our study due to the wide range of compounds we exposed them to. More than 160 of these ASE genes unidentified by GTEx showed evidence of cASE, thus pinpointing the environmental condition which was responsible for altering regulation of gene expression. Indeed a large fraction of CM cASE genes (>47%) are not eGenes in GTEx as detected by eQTL mapping or ASE. Because the CM samples were sequenced to very high depth, and because the power to detect allelic bias is based on read depth, this result could be partially due to having high statistical power to detect ASE and cASE. However, this does not seem to be the case, as we obtained similar results for LCLs and IPSCs, compared to the eQTL results in the Geuvadis and i2QTL datasets, respectively, with ASE genes being enriched in eGenes, but not cASE. While comparisons across studies may be complicated by several factors including differences in haplotype structures, study populations, and sequencing depth, the results are highly concordant and support the same conclusion.

Our results show the importance of considering the relevant tissue type, as CM-specific cASE are enriched for heart rate, which is a phenotype with a direct physiological role for the heart muscle. The importance of the identification of G×E in a range of environments is illustrated by our findings that Mendelian cardiovascular disease genes and TWAS genes in a wide variety of traits display G×E. In total, across all cardiovascular related traits we identified 169 genes with cASE, spanning the 12 conditions tested. In cardiomyocytes, 52 genes with cASE were associated with cardiovascular disease traits. Among the treatment effects, metal ions (cadmium and copper) and corticosteroids (dexamethasone) are the most common environments interacting with genetic risk for CVD and atrial fibrillation, respectively.

For example, the aryl hydrocarbon receptor nuclear translocator two gene (*ARNT2*) was identified by GWAS for atrial fibrillation (*Nielsen et al., 2018*) and is a treatment cASE gene in CMs for insulin. ARNT2 is part of the hypoxia-inducible factor (HIF) pathway (*Mandl et al., 2016*), which has a role in the progression of obesity and metabolic disease (*Gaspar and Velloso, 2018*). Indeed, insulin itself is known to induce the expression of *HIF1A* (*Treins et al., 2002*), which we confirm in our differential gene expression data. In turn, the HIF pathway is especially important for cardiovascular disease and the response to cardiac ischemia (*Semenza, 2014*). This approach could be further expanded by targeting individuals with large polygenic risk scores to discover additional G×E effects of clinical relevance, in future studies.

The polygenic nature of human complex traits provides a formidable challenge to tackle the genetic and molecular basis of interindividual variation. Our study demonstrates that interactions between genetic and environmental factors are common, but require specifically designed studies to be discovered. These results also have direct implications on issues related to the portability and interpretation of polygenic risk scores across individuals exposed to different environments. While gene-environment interactions further complicate the overall picture of human complex trait variation, they may represent an important contribution to the overall missing heritability that requires further study and careful consideration.

# Materials and methods

**Key resources table**

| Reagent type (species) or resource | Designation | Source or reference | Identifiers | Additional information |
|---|---|---|---|---|
| Cell line (*H. sapiens*, Female) | GM18858 | Coriell | | |
| Cell line (*H. sapiens*, Female) | GM18855 | Coriell | | |
| Cell line (*H. sapiens*, Female) | GM18505 | Coriell | | |
| Cell line (*H. sapiens*, Female) | GM18912 | Coriell | | |
| Cell line (*H. sapiens*, Female) | GM18520 | Coriell | | |
| Cell line (*H. sapiens*, Female) | GM19209 | Coriell | | |
| Cell line (*H. sapiens*, Female) | 18858_IPSC | Banovich2018 | | |
| Cell line (*H. sapiens*, Female) | 18855_IPSC | Banovich2018 | | |
| Cell line (*H. sapiens*, Female) | 18505_IPSC | Banovich2018 | | |
| Cell line (*H. sapiens*, Female) | 18912_IPSC | Banovich2018 | | |
| Cell line (*H. sapiens*, Female) | 18520_IPSC | Banovich2018 | | |
| Cell line (*H. sapiens*, Female) | 19209_IPSC | Banovich2018 | | |

## Cell culture and treatments

Experiments were conducted using three cell types derived from the same six individuals: Lymphoblastoid Cell Lines (LCL), Induced Pluripotent Stem Cells, (IPSC) and IPSC-derived Cardiomyocytes (CM). All cells were cultured at 37°C with 5% $CO_2$. Each batch consisted of the same cell type from three individuals, with 28 treatments and two controls. Each experiment was performed in duplicate for a total of 12 batches (*Figure 1*).

LCLs from six Yoruba individuals were purchased from Coriell: GM18858, GM18855, GM18505, GM18912, GM18520, and GM19209. LCLs were maintained at a density of 200,000 to 1 million cells/ml in supplemented RPMI media (500 mL RPMI-1640 with glutamine [Fisher Scientific, 15-040-CM], 75 mL FBS [Genemate S1200-500], 5 mL GlutaMAX [35050-061, ThermoFisher Scientific] and 5 mL penicillin/streptomycin). A total of 50,000 cells were plated per well of a round-bottom 96-well plate in 100 ul supplemented RPMI media 48 hr before treatment. Cells from each individual were plated in 32 wells representing 28 treatments and two controls.

Each of the six LCLs were reprogrammed into iPSCs using episomal reprogramming vectors and expanded on a layer of MEF prior to conversion to feeder-independent growth as previously described (*Banovich et al., 2018*). iPSCs were seeded on plates coated with a 1:100 dilution of Matrigel hESC-qualified Matrix (354277, Corning, Bedford, MA, USA) and cultured in iE8 media (Invitrogen A1517001) supplemented with penicillin/streptomycin. Cells were passaged every 3–5 days using dissociation reagent (0.5 mM EDTA, 300 mM NaCl in PBS). During plating of cells, media was supplemented with 10 uM ROCK inhibitor (stemolecule Y27632, stemgent 04-0012) to aid in cell adherence. Media was changed every day thereafter. 50,000 cells were plated per well of a Matrigel-coated flat-bottom 96-well plate in supplemented E8 media 48 hours before treatment. Cells from each individual were plated in 32 wells representing 28 treatments and two controls performed in duplicate. Additional information on IPSC reprogramming, including number of passages and differentiation batch can be found in LABEL:SuppTable15.

CMs were differentiated from iPSCs using small molecules as previously described (*Banovich et al., 2018*). Briefly, Wnt signaling was modulated by treating iPSCs with 12 μM of the GSK3 inhibitor CHIR99021 (4953, Tocris Bioscience, Bristol, UK) followed by 2 μM of the Wnt signaling inhibitor Wnt-C59 (5148, Tocris Bioscience). Cells start to spontaneously beat between days 7 and 10. The cardiomyocyte population was enriched through metabolic purification by culturing the cells in glucose-free, lactate-containing media from days 14 to 20. In order to promote aerobic

respiration following purification, on Day 20, the cell culture media was replaced with galactose-containing media (500mL DMEM [without glucose, Life Tech A11430-01], 50 mL FBS [GeneMate S1200-500], 5 mL sodium pyruvate [Gibco 11360-070], 2.5 mL HEPES [Fisher SH3023701], 5 mL GlutaMAX [35050-061, ThermoFisher], 5 mL penicillin/streptomycin and 990 mg galactose [Sigma G5388]). On Day 25, 5 days before treatment, cells were trypsinized and split to 114,000 cells/well in 100 µL of galactose-containing media on a Matrigel-coated flat-bottom 96-well plate. Cells from each individual were plated in 32 wells representing 28 treatments and 2 controls. Purity of the cell cultures was determined by measuring the expression of the cardiomyocyte-specific marker cardiac troponin T (564767, BD Biosciences) by flow cytometry. Flow cytometry was performed on Day 25 for 18912, 18520, and 19209 and on Day 27 for 18858, 18855, and 18505. Purities ranged from 44% to 95% (LABEL:SuppTable14). Additional information on CM differentiation, including cell counts and the day at which the cells spontaneously started beating can be found in LABEL:-SuppTable16. All lines tested negative for mycoplasma contamination. All cell lines were authenticated by genotyping.

Samples were treated on a total of twelve plates, and each plate was processed as a batch. Each plate contained samples from three individuals from a single cell type, exposed to 28 treatments plus two vehicle controls (water and ethanol, referred as Control 1 and Control 2). Additionally, each plate had a technical replicate performed the following day. Importantly, all three cell types for the same group of individuals were treated and harvested in parallel on the same day, to avoid that cell-type effects are confounded with batch effects. Cells were treated for six hours at concentrations listed in LABEL:SuppTable1. As in *Moyerbrailean et al., 2016*, concentrations were chosen based on the highest physiological concentrations as reported by the Mayo Clinic (http://www.mayomedicallaboratories.com) or the CDC (http://www.cdc.gov/biomonitoring/), as available.

## RNA library preparation

For RNA library preparation, each plate was prepared as a batch. Treated cells were collected by centrifugation at 2000 rpm and washed 2x using ice cold PBS. Collected pellets were lysed on the plate, using Lysis/Binding Buffer (Invitrogen), and frozen at −80°C. Poly-adenylated mRNAs were subsequently isolated from thawed lysates using the Dynabeads mRNA Direct Kit (Ambion) and following the manufacturer instructions. RNA-seq libraries were prepared using a protocol modified from the NEBNext Ultra II Directional RNA library preparation protocol to use 96 Barcodes from BIOOScientific added by ligation, as described in *Moyerbrailean et al., 2015*. Libraries from the same plate were pooled together and quantified using the KAPA Library Quantification Kit, following the manufacturer instructions and using a custom-made series of standards obtained from serial dilutions of the phi-X DNA (Illumina). Library pools were sequenced to an average of 9.5M 75bp PE reads. Within each pool, individual library concentrations were normalized and repooled to achieve comparable sequencing depths. Twelve treatments were selected for deep sequencing on the basis of the strong transcriptional response they provoked in at least one cell type, in addition to both controls. Samples selected for deep sequencing were pooled within each plate and further sequenced on the NovaSeq 6000 using 300bp PE reads. Each plate of LCLs and IPSCs were sequenced once on one lane for an average of 147M reads, and each plate of CMs was sequenced twice on one lane for an average of 273M reads. The number of reads per library can be seen in LABEL:SuppTable13.

## Alignment of RNA-sequencing

RNA-sequencing reads were aligned to the human reference genome using HISAT2 (*Kim et al., 2015*) (https://ccb.jhu.edu/software/hisat2/index.shtml, version hisat2-2.0.4) with the following options:

HISAT2 -x <genome> −1 <fastq_R1.gz> −2 <fastq_R2.gz> where <genome> represents the location of the genome file (genome_snp_tran), and <fastqs_R1.gz> and <fastqs_R2.gz> represent that sample's fastq files.

Multiple sequencing runs were merged for each sample using samtools (version 2.25.0). We removed PCR duplicates and further removed reads with a quality score of <10 (equating to reads mapped to multiple locations). Shallow and deep RNA-sequencing reads were aligned in an identical manner except for the reference used. GRCh38 was used for the deeply sequenced data and GRCh37 for the low coverage data which is not used after the initial screening.

## Differential gene expression analysis

To identify differentially expressed (DE) genes in shallow and deep sequencing data, we used DESeq2 (*Love et al., 2014*) (R version 3.5.2, DESeq2 version 1.28.1). coverageBed was utilized to count reads in transcripts from the Ensembl gene annotation with -s to account for strandedness and -split for BED12 input. The counts were then utilized in DESeq2 to determine changes in gene expression under the different treatment conditions.

$$\text{Model 1}: \text{CellType} + \text{Treatment.ID} \tag{1}$$

Each treatment was compared to its relevant vehicle control, except for plate CM1R2, where we identified a technical problem with control 1, so we only used control two for this plate (LABEL:SuppTable13). Note that comparison between the two controls yielded the expected finding of fewer than six differentially expressed genes for all other plates, thus confirming that the two controls are essentially technical replicates of the untreated condition. Multiple test correction was performed using the Benjamini-Hochberg procedure (*Benjamini and Hochberg, 1995*) with a significance threshold of 10%. A gene was considered a DEG if at least one of its transcripts was differentially expressed and had an absolute $log_2$ fold change >0.25. Each cell type was run independently, and the model corrected for a composite variable of library preparation batch and individual. Full DESeq2 results from the shallow sequencing can be found in LABEL:SuppTable7.

After identifying DEGs from all 28 treatments in the initial shallow sequencing, we selected 12 to sequence more deeply. Nine of these twelve treatments were selected because they resulted in at least 60 DEGs in CMs. Selenium and Zinc were added because they induced a strong response in both IPSCs and LCLs, and Cadmium was added to complete the set of metal treatments. Full DESeq2 results from the deep sequencing can be found in LABEL:SuppTable8.

To identify genes that show evidence of treatment × cell type interactions, we analyzed the deep sequencing data across all cell types and treatments using the following likelihood ratio test to compare Model 1 and Model 2:

$$\text{Model 2}: \text{CellType} + \text{Treatment.ID} + \text{CellType}:\text{Treatment.ID} \tag{2}$$

## Differential intron excision analysis

To detect shifts in splicing patterns across cell types and environments we used LeafCutter (*Li et al., 2018*), an intron-based splicing analysis tool. Briefly, LeafCutter uses short RNA-seq reads spanning exon-exon junctions (i.e. split reads) to estimate internal introns usage, hence being able to ultimately infer all splicing events which can be summarized with differential introns excisions. Overlapping introns sharing a splice sites are then identified, which are subsequently used to construct a graph where the connected components represent clusters. Lastly, for each cluster, the counts of the composing introns are jointly modeled with a Dirichlet-multinomial generalized linear model. We used the provided bam2junc.sh script to convert HISAT2-generated .bam files from the deep sequencing data to .junc files, as well as leafcutter_cluster.py script to perform intron clustering with the -s yes -m 50 -l 500000 options (keep strand information, >50 split reads supporting each cluster and accepting introns with length up to 500kb). We removed clusters localizing on sex, mitochondrial and scaffold chromosomes, and used leafcutter_ds.R to perform differential intron excision analysis by contrasting each treatment with the appropriate control, with individuals (i.e. cell lines) as confounders.

## Determining significance of global directional shifts in RNA processing

LeafCutter produces an output file in which, for every intron within a successfully tested cluster it computes the percentage spliced in index ($\Psi$) in the control and treatment condition, as well as the corresponding $\Delta\Psi$. LeafCutter exploits that retention and alternative excision of introns (*Braunschweig et al., 2014*; *Boutz et al., 2015*) act as proxies of several subtypes of alternative splicing events, including exon skipping (ES), intron retention (IR), mutually exclusive exons (MxE), alternative first and last exons (AFE/ALE), alternative 5' and 3' splice sites (A5SS/A3SS), among others. For every condition, we extracted introns belonging to clusters considered to be differentially spliced (FDR < 10%), and we performed a two-sided binomial test defining n as the number of introns with $\Delta\Psi > 0$, and p equal to the average proportion of positive events among all conditions. We considered a shift to be significant if the corresponding FDR < 10%.

## Gene ontology, KEGG, disease-gene network enrichment analyses

We used ClusterProfiler (*Yu et al., 2012*) R package to perform Gene Ontology (*Ashburner et al., 2000*; *The Gene Ontology Consortium, 2019*), KEGG (*Wixon and Kell, 2000*), and Disease-Gene Network (*Piñero et al., 2020*) enrichment analyses. We considered two separate approaches for enrichment analysis. First, we calculated an enrichment of DEGs within each treatment per cell type against a background of all expressed genes in that treatment. This identified the pathways that were most enriched in DEGs for each treatment (LABEL:SuppTable2). Second, we calculated an enrichment within each cell type across all environments to compare GO, KEGG and DGN enrichments using the compareCluster function by submitting the lists of differentially spliced or expressed (FDR < 10%) genes in each condition, as well as one background gene list comprising all tested genes in that given cell type. We considered a process to be significantly enriched if FDR < 10%.

For performing GO analysis on ASE variance, we first identified the top 20% and bottom 20% of genes based on their variance (high variance or low variance genes, respectively). We tested for pathway enrichments using Gene Ontology annotations comparing the high variance against the low variance genes using ClusterProfiler with the same thresholds as described above (LABEL:SuppTable6).

## Variance partitioning of gene expression and intron excision

We used the Variance Partition package (*Hoffman and Schadt, 2016*) to partition the total variance in gene expression. For each gene, the tool determines the fraction of the observed variance explained by each variable by implementing a linear-mixed model, ultimately allowing multiple dimensions of variation to be analyzed simultaneously. Variance which cannot be attributed to any of the provided covariates is counted as residuals, which can be generally interpreted as due to noise that can have technical or biological origin. We included all transcripts with at least one count per million reads (cpm) in 50% of the libraries, and the resulting data was voom-normalized (*Ritchie et al., 2015*; *Law et al., 2014*). All variables included in the model were categorical and were therefore modeled as random effects. For the model including all cell types, we used the fitExtractVarPartModel function with the following formula: (1|Individual) + (1|CellType) + (1|Plate) + (1|Treatment). When each cell type was considered independently, we dropped CellType and added a Individual:Treatment variable, accounting for the interaction between these two covariates. We restricted this analysis to genes tested for both splicing and gene expression so we can compare the origins of variation in these two transcriptional processes.

For splicing, we merged all LeafCutter .junc files across every cell type and condition, where each column represents a sample and each row a different intron, in order to recover the full list of detected introns. We applied the suggested cutoff to extract introns considered to be expressed (on average, at least one cpm in half of the libraries), and used the cell type covariate to design the matrix eventually used by voom to normalize expression counts. Finally, we used the same methods as for gene expression to partition the variance.

## Genotyping and ASE quantification with QuASAR

Using the common SNPs from dbSNP Build 144, we calculated the number of RNA-seq reads mapping to each allele using samtools mpileup. We inferred genotypes from these pileups using all samples from a given individual across all treatments and each plate together using QuASAR (*Harvey et al., 2015*). Counts for each pair of controls on the same plate were combined. For all subsequent analyses, we focused on heterozygous SNPs with a read coverage greater than five and located on autosomes. Initial testing for ASE was performed for each sample separately using QuASAR. The amount of ASE identified via QuASAR aggregated by treatment is shown in *Figure 3A*, and the full output from QuASAR for tested SNPs can be found in LABEL:SuppTable3.

## ANOVA for identifying ASE

Previous efforts for detecting ASE have commonly used binomial or beta-binomial tests (*Harvey et al., 2015*; *Knowles et al., 2017*; *Degner et al., 2012*). Given that we have a large number of replicates and experimental conditions and to avoid excluding conditions where an allele may be lowly expressed, we tested for ASE using a linear model which incorporated ASE measurements across all conditions for a SNP in a given individual. To this end, we re-calculated ASE by adding a pseudocount to both reference and alternate read counts, and then computed the natural logarithm of the number

of reference reads divided by the number of alternate reads. By doing so, we directly infer the noise inherent in ASE measurements using the linear model, obviating the need for binomial models to quantify the technical variance. To detect ASE in the model with all cell types combined, we selected 69,683 SNPs for which ASE was measured in at least five conditions per cell type, including at least one measurement for each control. We used an ANOVA test to compare a full model including the control ID, cell type, treatment, and cell type × treatment interaction term against a reduced model with the intercept set to 0:

$$\text{Full}: \text{ASE} \sim \text{Control} + \text{CellType} + \text{Treatment} + \text{CellType} : \text{Treatment} \qquad (3)$$
$$\text{Reduced}: \text{ASE} \sim 0 \qquad (4)$$

We only consider this test for SNPs with a sufficient number of observations given the number of parameters being estimated (i.e., at least 5 degrees of freedom). Full results for the ANOVA model can be found in LABEL:SuppTable9. For SNPs with ASE detected by the ANOVA model, we performed spearman correlations for every pair of libraries using all SNPs shared between them. Pairs of libraries were then placed into four categories based on their differences. These were: 'Across Cell' (n = 968 pairs), which included all pairs of libraries from the same individual and treatment, but different cell types; 'Across Individual' (n = 490 pairs), which included all pairs of libraries from the same treatment, cell type, and plate, but different individuals; 'Across Replicate' (n = 242 pairs), which included all pairs of libraries from the same individual, cell type, and treatment, but different plate; and 'Across Treatment' (n = 3146 pairs), which included all pairs of libraries from the same individual, cell type, and plate, but different treatment. Finally, the category for 'Control 1 vs Control 2' (n = 33 pairs) includes correlations between the two control libraries from the same individual, cell type, and plate.

## Fixed effect linear model for identifying conditional ASE

To identify SNPs displaying cASE due to cell type, treatment, or the interaction between cell type and treatment, we used a fixed effect linear model. All SNPs with significant ASE detected via the ANOVA model were tested for cASE. Additionally, SNPs must have ASE measured in at least one of each control for each cell type to be tested. We modeled the ASE values for each SNP in an individual across all conditions as a function of cell type, treatment, cell type-treatment interaction, and a variable representing the vehicle used for that treatment (Control):

$$\text{ASE} \sim \text{Control} + \text{CellType} + \text{Treatment} + \text{CellType} : \text{Treatment} \qquad (5)$$

Results can be found in LABEL:SuppTable10, and are summarized in LABEL:SuppTable_SigTestAll. We also investigate ASE, and cASE for each cell type separately which may recover situations in which a gene is only expressed in one cell type. First we detect ASE for each cell type across any condition:

$$\text{Full}: \text{ASE} \sim \text{Control} + \text{Plate} + \text{Treatment} \qquad (6)$$
$$\text{Reduced}: \text{ASE} \sim 0 \qquad (7)$$

Then, we test for each treatment component in the Full model to detect cASE. For the model with the cell types analyzed separately, ANOVA results can be found in LABEL:SuppTable_ANOVAsep, and results from the Full, fixed effect model (Equation 6), can be found in LABEL:SuppTable_FixedSep and are summarized in LABEL:SuppTable_SigTestSep.

## Genomic locations of ASE and cASE SNPs

Using the genomic annotations from dbSNP build 153, we assigned SNPs tested for ASE to one of 18 categories on the basis of their location in relation to a gene and predicted function. For each category, we calculated the enrichment of ASE and cASE SNPs separately relative to all SNPs tested for ASE and performed Fisher's exact test to test for significance. Multiple test correction of p-values was performed using the Benjamini-Hochberg procedure.

## cASE overlap with TWAS, OMIM, and other G×E studies

To identify traits which are enriched for CM cell type cASE, we focused on 8 TWAS for cardiovascular traits from *Zhang et al., 2020*. For each trait, we intersected all cell type cASE with TWAS genes, and calculated the proportion of overlap which is attributable to CM cell type cASE only. We used

the one-sample proportion test to determine if the CM cell type cASE overlap for a given trait was significantly different from the average trait overlap for CM cell type cASE.

To identify genes with treatment cASE that could influence complex traits, we intersected our cASE genes from the model with all cell types combined with genes whose expression has been implicated by transcriptome-wide association study (TWAS) to influence complex traits in 103 TWAS (*Zhang et al., 2020*). Forty-five TWAS had at least 20 cASE gene overlaps across all treatments, and those were selected for further analysis. As above for CM cell type cASE, we used the one-sample proportion test to determine if the treatment cASE overlap for a given trait was significantly different from the average overlap for that trait across all treatments.

$$P(t,c) = \frac{O(t,c)}{N(t)}$$

$$Z(t,c) = \frac{P(t,c) - P_0(c)}{\sqrt{P0(c) * (1 - P_0(c))/N(t)}}$$

where $t$ is the trait, $c$ is the treatment, $O(t,c)$ is the number of genes which are significant for both the trait and the treatment, $N(t)$ is the sum of all genes for the trait which are treatment cASE in any treatment, and $P_0(c)$ is the average $P(t,c)$ across all traits.

For plotting purposes, we abbreviated the TWAS study names from *Zhang et al., 2020* according to LABEL:SuppTable11.

To identify genes responsible for Mendelian traits, we downloaded OMIM's Synopsis of the Human Gene Map (morbidmap.txt) on December 19, 2019 and intersected treatment cASE genes from the model containing only the CM data with OMIM genes. This resulted in 95 OMIM traits, which were manually curated to identify traits relevant to heart disease.

To investigate whether our G×E results replicate in other environments previously investigated, we calculated the overlap between genes with gene × treatment interactions in our dataset and genes with G×E (P < 0.05) identified in fourteen previous ASE and eQTL mapping studies (*Maranville et al., 2011*; *Idaghdour et al., 2012*; *Mangravite et al., 2013*; *Çalışkan et al., 2015*; *Moyerbrailean et al., 2016*; *Quach et al., 2016*; *Zhernakova et al., 2017*; *Knowles et al., 2017*; *Leland Taylor et al., 2018*; *Huang et al., 2021*; *Knowles et al., 2018*; *Lee et al., 2014*; *Nédélec et al., 2016*; *Barreiro et al., 2012*). The full table of treatment cASE genes which replicated in these other studies can be found in LABEL:SuppTable4.

## Enrichment of ASE and cASE genes in GTEx eGenes

We used Fisher's exact test to test for an enrichment in ASE and cASE genes in eGenes from three large eQTL studies for CMs, IPSCs, and LCLs, respectively: GTEx left ventricle and atrial appendage (*The Gtex Consortium, 2020*), i2QTL (*Bonder et al., 2021*), and Geuvadis (*Lappalainen, 2013*; *Wen et al., 2015*). ASE and cASE genes for each cell type were detected using the model that examined each cell type separately. For each cell type, we restricted our analysis to include genes which had been tested for being eGenes in the relevant eQTL study and had also been tested for ASE or cASE in our study. GTEx v8 eQTL data was downloaded from https://storage.googleapis.com/gtex_analysis_v8/single_tissue_qtl_data/GTEx_Analysis_v8_eQTL.tar. GTEx ASE data were downloaded from https://github.com/secastel/phaser/blob/master/gtex_v8_analyses/gtex_v8_tissue_by_gene_imbalance.tar.gz. i2QTL eGenes and tested genes were found in Supplemental Tables 3 and 7 in *Bonder et al., 2021*. Geuvadis eGenes were downloaded from https://www.ebi.ac.uk/arrayexpress/experiments/E-GEUV-3/files/analysis_results/EUR373.gene.cis.FDR5.best.rs137.txt.gz. The list of tested Geuvadis genes were downloaded from http://www-personal.umich.edu/~xwen/geuvadis/geuv.fm.tar.gz.

## Heritability explained by cASE

To estimate the heritability explained by G×E, we used RHE-mc (*Pazokitoroudi et al., 2020*). We performed two separate analyses using the UK Biobank data. First, we quantified the heritability of complex traits using an annotation of CM gene × cell type genes. Second, we quantified the heritability of complex traits using an annotation of gene × treatment genes. In both cases, SNPs were annotated to G×E genes within 100 Kb. For the gene × treatment analysis, we included both gene × treatment and gene × treatment × cell type genes, with each treatment forming a separate annotation.

The heritability partitioning was performed as described in *Pazokitoroudi et al., 2020*. Briefly, we excluded SNPs with greater than 1% missingness and minor allele frequency smaller than 0.1%. Further, we excluded SNPs that fail the Hardy-Weinberg test at significance threshold $10^{-7}$ as well as SNPs that lie within the MHC region (Chr6: 25–35 Mb) to obtain 7, 774, 235 SNPs. We included age, sex, and the top 20 genetic principal components (PCs) as covariates in our analysis for all traits. We used PCs precomputed by the UK Biobank from a superset of 488, 295 individuals. Additional covariates were used for waist-to-hip ratio (adjusted for BMI) and diastolic/systolic blood pressure (adjusted for cholesterol-lowering medication, blood pressure medication,insulin, hormone replacement therapy, and oral contraceptives). For the CM gene × cell type annotation, we partitioned the heritability of 17 complex traits, while for the gene × treatment analysis, we expanded to 22 traits in the same sample from the UK Biobank. For each annotation, we computed the heritability enrichment as the ratio of the percentage of heritability explained to the percentage of SNPs in that annotation.

## Mixed effect linear model for quantifying ASE variance

To quantify ASE variance, we used the same linear model as for identifying cASE, with the exception of modeling the cell type, treatment, cell type-treatment interaction, and batch variables as random effects:

$$\text{ASE} \sim (1\,|\,\text{Control}) + (1\,|\,\text{CellType}) + (1\,|\,\text{Treatment}) + (1\,|\,\text{CellType} : \text{Treatment}) \tag{8}$$

As before, we only tested SNPs with significant ASE as determined by the ANOVA. To analyze each cell type independently, we used the following model:

$$\text{ASE} \sim (1\,|\,\text{Control}) + (1\,|\,\text{Batch}) + (1\,|\,\text{Treatment}) \tag{9}$$

## Biological features of ASE variance

Residual variance is partially a function of sequencing depth, so for all analyses of variance components for ASE, at each SNP, we have adjusted for the total number of reads covering that position. To determine the biological significance of ASE variance, we identified biological features which contributed to differences in treatment or residual variance. Given the overwhelming contribution of cell type to ASE variance, we used the ASE variance calculated within each cell type. We tested for a relationship between residual and treatment variance and six annotations using the following model:

$$\text{Variance} \sim \text{Annotation} + \text{SNP expression} \tag{10}$$

We assigned SNPs tested for ASE to the genes in which they reside, and considered the following sevengene annotations:

1. Differentially expressed genes: We categorized genes as (1) not differentially expressed in any condition; (2) differentially expressed in at least one condition (absolute value of logFC > 0.25 but < 0.75); or (3) highly differentially expressed (absolute value of logFC > 0.75) in at least one condition.
2. Differentially spliced genes: Genes were classified as being differentially spliced as described in the "Differential intron excision analysis" section.
3. Promoter architecture characteristics: Information on promoter characteristics, including presence of a TATA box, GC percentage, and CpG percentage were downloaded from refTSS (*Abugessaisa et al., 2019*): http://reftss.clst.riken.jp/datafiles/current/human/tata_box_annotations/hg38_tata_annotation_v3.txt.gz
4. Loss of function tolerance: Loss of function statistics by gene were downloaded from Gnomad, version 2.1.1 (https://storage.googleapis.com/gnomad-public/release/2.1.1/constraint/gnomad.v2.1.1.lof_metrics.by_gene.txt.bgz). Specifically, we use the observed over expected ratio for predicted loss of function variants for a transcript (oe_lof). Lower scores indicate less tolerance to loss of function mutations. For example, haploinsufficient genes tend to have low oe_lof scores (*Karczewski et al., 2020*).
5. dN/dS ratio: dN/dS ratios were downloaded from Lindblad-Toh et al (*Lindblad-Toh et al., 2011*). Ratios were derived from an analysis of 29 mammalian genomes. (http://www.broadinstitute.org/ftp/pub/assemblies/mammals/29mammals/PositivelySelectedCodons.tar.gz).
6. Cardiomyopathy genes: The 55 genes tested by the Mayo Clinic Laboratory Comprehensive Cardiomyopathy Multi-Gene Panel were considered as cardiomyopathy genes. For this category, we only tested ASE variance in CMs.

7. TWAS genes: All genes significant for at least one trait in the PTWAS analysis by (*Zhang et al., 2020*) were considered TWAS genes.

## Data and code availability

Sequencing files have been uploaded to the Sequence Read Archive (SRA) under Bioproject PRJNA694697. Code is available at https://github.com/piquelab/GxExC (*Findley, 2021*; copy archived at swh:1:rev:15df015227a05ce566fff158d312bd1a666e1235).

## Acknowledgements

We thank members of the Luca, Pique-Regi, and Gilad labs for helpful discussions and comments. We thank Tuuli Lapplainen and Stephane Castel for providing access to the GTEx ASE results. We thank the editors and reviewers for their thoughtful suggestions and review. This work was supported by the National Institute of General Medical Sciences of the National Institutes of Health (R01GM109215 to FL and RP, R35GM131726 to YG., F30GM131580 to ASF, and R35GM125055 to SS) and NSF grants III-1705121 to AP. and SS. This research was conducted using the UK Biobank Resource under application 33127. We thank the participants of UK Biobank for making this work possible.

## Additional information

### Funding

| Funder | Grant reference number | Author |
|---|---|---|
| National Institute of General Medical Sciences | R01GM109215 | Roger Pique-Regi Francesca Luca |
| National Institute of General Medical Sciences | R35GM131726 | Yoav Gilad |
| National Institute of General Medical Sciences | F30GM131580 | Anthony S Findley |
| National Institute of General Medical Sciences | R35GM125055 | Sriram Sankararaman |
| National Science Foundation | III-1705121 | Ali Pazokitoroudi Sriram Sankararaman |

The funders had no role in study design, data collection and interpretation, or the decision to submit the work for publication.

### Author contributions

Anthony S Findley, data-curation, formal-analysis, visualization, writing-original-draft, Writing – review and editing; Alan Monziani, formal-analysis, visualization, writing-original-draft, Writing – review and editing; Allison L Richards, Michelle C Ward, Investigation, Methodology, Writing – review and editing; Katherine Rhodes, Cynthia A Kalita, Investigation, Writing – review and editing; Adnan Alazizi, Investigation; Ali Pazokitoroudi, formal-analysis, Software, visualization, Writing – review and editing; Sriram Sankararaman, Xiaoquan Wen, Software, Supervision, Writing – review and editing; David E Lanfear, Writing – review and editing; Roger Pique-Regi, Francesca Luca, conceptualization, funding-acquisition, Methodology, project-administration, resources, Software, Supervision, writing-original-draft, Writing – review and editing; Yoav Gilad, conceptualization, funding-acquisition, Methodology, project-administration, resources, Supervision, Writing – review and editing

### Author ORCIDs

Anthony S Findley http://orcid.org/0000-0001-9922-3076
Michelle C Ward http://orcid.org/0000-0003-1485-320X
Roger Pique-Regi http://orcid.org/0000-0002-1262-2275
Yoav Gilad http://orcid.org/0000-0001-8284-8926
Francesca Luca http://orcid.org/0000-0001-8252-9052

Decision letter and Author response
Decision letter https://doi.org/10.7554/eLife.67077.sa1
Author response https://doi.org/10.7554/eLife.67077.sa2

## Additional files

### Supplementary files
• Supplementary file 1. Supplementary Tables.
• Transparent reporting form

### Data availability
Sequencing files have been uploaded to the Sequence Read Archive (SRA) under Bioproject PRJNA694697.

The following dataset was generated:

| Author(s) | Year | Dataset title | Dataset URL | Database and Identifier |
|---|---|---|---|---|
| Findley A, Pique-Regi R, Gilad Y, Luca F | 2021 | | http://www.ncbi.nlm.nih.gov/bioproject/694697 | NCBI BioProject, PRJNA694697 |

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
