## [Decision Letter]

**Acceptance summary:**

This manuscript is of broad interests to readers in the area of complex trait genetics. It uses an established experimental design (profiling of allele-specific gene expression changes) to demonstrate the extensive role of gene-environment interactions in cell-type-specific gene expression regulation. We believe this is important work in the area of context-specific genetic regulation of gene expression.

**Decision letter after peer review:**

Thank you for submitting your article "Functional dynamic genetic effects on gene regulation are specific to particular cell types and environmental conditions" for consideration by *eLife*. Your article has been reviewed by 2 peer reviewers, one of whom is a member of our Board of Reviewing Editors, and the evaluation has been overseen by Patricia Wittkopp as the Senior Editor. The following individual involved in review of your submission has agreed to reveal their identity: Kaur Alasoo (Reviewer #2).

Essential revisions:

1. In your analysis of ASE overlap with eQTLs, you only focussed on cardiomyocytes and the heart tissue from GTEx. While this analysis is solid, it raises some issues around how well are your cardiomyocytes matched to the heart tissue in GTEx and how well does your claim that 47% of the genes with dynamic regulatory interactions are missed in existing large eQTL datasets generalise to the other two cell types in your study (LCLs and IPSCs). Fortunately, high-resolution eQTL maps for the other two cell types (iPSCs and LCLs ) do exist, so it should be possible to check this.

2. The authors show that their cASE catalogue captures genes that are missed by GTEx and identifies potentially novel GWAS genes. For these analyses, they focus mostly on the CM data where it's not clear that GTEx has a directly comparable cell type and ASE signal is not the same signal GTEx is generating. Because measuring ASE is not the same as measuring eQTL, such a comparison might not be fair due to statistical power and MAF differences. It could be better to compare ASE within the GTEx data. Also, the CM pool was sequenced to very high depth, and because the power to detect allelic bias is based on read depth, this study should have high power. I wonder how many of the (c)ASE they report (that's not observed in GTEx) is just a reflection of the high statistical power in this study due to read depth.

3. You report that 55% of the variance in gene expression in cardiomyocytes was assigned to differences between individuals, whereas this was much lower in LCLs (36%) and iPSC (28%). Could this simply reflect the fact that in your experimental design, genetic differences between individuals are confounded by CM differentiation batch and CM differentiation just happens to be highly variable? As far as I understood, you only performed one differentiation from each individual and thus you are not able to separate out the effect of differentiation batch from the effect of the individual?

*Reviewer #1:*

This manuscript is from an exceptional team of investigators. They used a well-reasoned two-stage study design to quantify the genetic effects on gene expression in the context of different environments. They studied six individuals, three cell types (LCC, iPSC, iPSC->CM), and 28 different treatment conditions. They then performed an initial stage of relatively shallow RNA-seq (median 9.5M reads per sample), identified DE genes, then selected 12 conditions which had a large global change to the transcriptome. In the second stage, they sequenced the RNA from these conditions to much higher depth (median 146M LCL, 148M iPSC, 273M CMc), which allows for the quantification of allelic bias at heterozygous positions. The authors then compare the pattern of response-specific allelic expression to make several biological inferences.

The authors show that their cASE catalogue captures genes that are missed by GTEx and identifies potentially novel GWAS genes. For these analyses, they focus mostly on the CM data where it's not clear that GTEx has a directly comparable cell type and ASE signal is not the same signal GTEx is generating. Because measuring ASE is not the same as measuring eQTL, such a comparison might not be fair due to statistical power and MAF differences. It could be better to compare ASE within the GTEx data. Also, the CM pool was sequenced to very high depth, and because the power to detect allelic bias is based on read depth, this study should have high power. I wonder how many of the (c)ASE they report (that's not observed in GTEx) is just a reflection of the high statistical power in this study due to read depth. If the increased detection is just due to the technical property of read depth, this is less interesting. However, overall this is a nice study that highlights the importance of environmental context to the genetic control of gene expression.

The authors quantify replication of their cASE signals with 14 previous ASE and eQTL mapping studies that measured response. They report nominal replication overlaps here, but it's not clear if there's any enrichment for reproducibility. Is what they report more than expected?

Some of the GWAS comparisons are unclear. In one paragraph, the authors compare their results to gene sets that come from PTWAS that come from GTEx eQTL applied to GWAS data. Here, the target genes seem like a reasonable set. Later, the authors mention GWAS genes as those nearby GWAS signals, without PTWAS or colocalization support – I think. This gene set is more problematic in that the effector transcript at the locus is not as clear.

*Reviewer #2:*

In this study, Findley et al. profile gene expression and splicing in three cell types (lymphoblastoid cell lines, induced pluripotent cells and induced cardiomyocytes), six individuals and 28 treatments (12 of which were sequenced deeply). In addition to characterising the changes in gene expression and splicing between cell types and in response to various treatments, they were also able to use allele-specific expression (ASE) analysis to quantify the role of genetic variation in the regulation of gene expression in these cell types and contexts. They identify allele-specific expression in a large number of genes and demonstrated that many of these effects are specific to a subset of the cell types and/or stimulations. Importantly, these effects are often missed in large-scale eQTL studies such as GTEx, because they usually profile cells and tissues only in one baseline condition. These results highlight that larger eQTL maps or novel experimental techniques are needed to pinpoint the functional consequences of these non-coding genetic variants.

Strengths

A major strength of this study is that all three cell types originated from the same set of six unrelated individuals. This ensured that observed differences in allele-specific expression between cell types were not confounded by genetic differences between individuals, as the same set of genes contained the same set of heterozygous genetic variants in all three cell types. Without this design choice, it would not have been possible to accurately quantify the extent of cell-type-specific ASE.

Another strength of the study is the two-step design where the authors first used shallow RNA-seq to profile 28 different treatments and subsequently selected 12 of those with the largest effects for high-coverage RNA-seq. This was important because reliably detecting ASE requires much higher read coverage than simply detecting differential expression (enough reads need to capture both alleles of the heterozygous variants).

The analyses in the manuscript have been carefully conducted using established methods. I believe the choice of methods to be well justified and I don't have any major concerns here.

Weaknesses

In your analysis of ASE overlap with eQTLs, you only focussed on cardiomyocytes and the heart tissue from GTEx. While this analysis is solid, it raises some issues around how well are your cardiomyocytes matched to the heart tissue in GTEx and how well does your claim that 47% of the genes with dynamic regulatory interactions are missed in existing large eQTL datasets generalise to the other two cell types in your study (LCLs and IPSCs). Fortunately, high-resolution eQTL maps for the other two cell types (iPSCs and LCLs ) do exist, so it should be possible to check this.

You report that 55% of the variance in gene expression in cardiomyocytes was assigned to differences between individuals, whereas this was much lower in LCLs (36%) and iPSC (28%). Could this simply reflect the fact that in your experimental design, genetic differences between individuals are confounded by CM differentiation batch and CM differentiation just happens to be highly variable? As far as I understood, you only performed one differentiation from each individual and thus you are not able to separate out the effect of differentiation batch from the effect of the individual?

It would have been nice to see the impact of genetic variants quantified at the level of allele-specific splicing, but I completely understand that this can be tricky (if not impossible) to do using current short-read sequencing approaches and the accompanying alignment issues that can disproportionally affect reads originating from exon-exon junctions. Nevertheless, it would be nice to see some discussion about what would be needed for the field to quantify allele-specific splicing changes. I suspect also that sequencing full-length transcripts with long-read technology could be helpful.

The eQTL Catalogue contains uniformly processed eQTL summary statistics both from the HipSci (iPSCs) and GEUVADIS (LCLs) studies that you could use to expand your eQTL overlap analysis to the other two cell types and see how well your results generalise. Furthermore, since iPSC and LCL cultures are usually highly pure, you would not need to worry as much about differences in cell-type composition between your CM cultures and GTEx heart tissue.

---

## [Author Response]

Essential revisions:1. In your analysis of ASE overlap with eQTLs, you only focussed on cardiomyocytes and the heart tissue from GTEx. While this analysis is solid, it raises some issues around how well are your cardiomyocytes matched to the heart tissue in GTEx and how well does your claim that 47% of the genes with dynamic regulatory interactions are missed in existing large eQTL datasets generalise to the other two cell types in your study (LCLs and IPSCs). Fortunately, high-resolution eQTL maps for the other two cell types (iPSCs and LCLs ) do exist, so it should be possible to check this.

We thank the reviewers for this excellent suggestion. We have now added comparisons to eQTL results also for LCLs and IPSCs. For LCLs, we compared our ASE and cASE results to the eQTLs from the GEUVADIS dataset, for the IPSCs we have made a comparison to the eQTLs from the I2QTLs consortium. For both cell types, we find similar results to what we observed from the CMs, with a 1.5-fold enrichment for the ASE but no significant enrichment for the cASE. We have now added these results in figure 3E and we have added the following text to the manuscript:

Abstract

"On average half of the genes with dynamic regulatory interactions were missed by large eQTL mapping studies, indicating the importance of exploring multiple treatments to reveal previously unrecognized regulatory loci that may be important for disease."

Results

"We next investigated whether these genetic effects on gene expression have been previously observed in large scale eQTL mapping studies that largely ignored dynamic regulatory interactions. […] In IPSCs, there were 3,113 genes with ASE, and 80% were eGenes in i2QTL (1.49-fold enrichment, p = 1.0 x 10^-10^). Of the 352 genes with cASE in IPSCs, 284 were eGenes in i2QTL. As with the CMs, these cASE genes were not significantly enriched (odds ratio = 1.03)."

Discussion

"This is reflected in the lack of enrichment for cASE genes in eQTLs from large studies in three tissues/cell-types. […] However, this does not seem to be the case, as we obtained similar results for LCLs and IPSCs, compared to the eQTL results in the Geuvadis and i2QTL datasets, respectively, with ASE genes being enriched in eGenes, but not cASE."

Methods

"We used Fisher's exact test to test for an enrichment in ASE and cASE genes in eGenes from three large eQTL studies for CMs, IPSCs, and LCLs, respectively: GTEx left ventricle and atrial appendage (Aguet et al., 2020), i2QTL (Bonder et al., 2021), and Geuvadis (Lappalainen et al. 2013, Wen et al., 2015). […] Geuvadis eGenes were downloaded from https://www.ebi.ac.uk/arrayexpress/experiments/E-GEUV-3/files/analysis_results/EUR373.gene.cis.FDR5.best.rs137.txt.gz. The list of tested Geuvadis genes were downloaded from http://www-personal.umich.edu/~xwen/geuvadis/geuv.fm.tar.gz."

2. The authors show that their cASE catalogue captures genes that are missed by GTEx and identifies potentially novel GWAS genes. For these analyses, they focus mostly on the CM data where it's not clear that GTEx has a directly comparable cell type and ASE signal is not the same signal GTEx is generating. Because measuring ASE is not the same as measuring eQTL, such a comparison might not be fair due to statistical power and MAF differences. It could be better to compare ASE within the GTEx data. Also, the CM pool was sequenced to very high depth, and because the power to detect allelic bias is based on read depth, this study should have high power. I wonder how many of the (c)ASE they report (that's not observed in GTEx) is just a reflection of the high statistical power in this study due to read depth.

We agree with the reviewers that it would be interesting to compare our results to the ASE analysis in the GTEx dataset. We requested access to the GTEx ASE results per gene/tissue (Castel et al., 2020) which are now publicly available, and made this comparison for the CM ASE and the GTEx heart tissues ASE. We found similar results to the ones originally reported for the eGenes (see Figure 3E and specific changes below).

To investigate whether the additional cASE that we find and were not detected in GTEx could be partially explained by our high statistical power in the CMs, and also to do a direct comparison across the same cell types, we repeated the same analysis in the LCLs and IPSCs, which were sequenced at lower depth (146M and 148M, compared to 273M in CMs). Our new results (see response to comment above) show that even when considering LCLs and IPSCs, we can still identify 141 and 68 genes with cASE that were missed in eQTL studies in LCLs (Geuvadis) and IPSCs (i2QTL), respectively. We have added the results of these analyses in figure 3E and the relevant changes to the manuscript are reported below.

Results

"In addition to eQTL mapping, GTEx also used ASE to measure *cis*-regulatory effects (Castel et al., 2020). As with GTEx eGenes, GTEx left ventricle and atrial appendage genes with ASE were enriched for CM ASE genes (odds ratio = 1.75, 1.66 for each tissue, respectively, p<10^-16^), but not CM cASE genes."

Discussion

"Indeed a large fraction of CM cASE genes (>47%) are not eGenes in GTEx as detected by eQTL mapping or ASE. […] While comparisons across studies may be complicated by several factors including differences in haplotype structures, study populations, and sequencing depth, the results are highly concordant and support the same conclusion."

Methods

"GTEx ASE data were downloaded from https://github.com/secastel/phaser/blob/master/gtex_v8_analyses/gtex_v8_tissue_by_gene_imbalance.tar.gz. i2QTL eGenes and tested genes were found in Supplementary File 1c and Supplementary File 1g (Supplemental Tables 3 and 7 in version with tracked changes) in Bonder et al., 2021. […] The list of tested Geuvadis genes were downloaded from http://www-personal.umich.edu/~xwen/geuvadis/geuv.fm.tar.gz."

3. You report that 55% of the variance in gene expression in cardiomyocytes was assigned to differences between individuals, whereas this was much lower in LCLs (36%) and iPSC (28%). Could this simply reflect the fact that in your experimental design, genetic differences between individuals are confounded by CM differentiation batch and CM differentiation just happens to be highly variable? As far as I understood, you only performed one differentiation from each individual and thus you are not able to separate out the effect of differentiation batch from the effect of the individual?

We agree with the reviewers that besides genetics, other factors may also contribute to the individual component of the variance in gene expression. To specifically address the question of a contribution from the differentiation process, we considered the expression of the gene TNNT2 which encodes for the Cardiac muscle troponin T. The expression of this gene is used as a marker of differentiation of CMs and a surrogate of CM purity. We repeated the analysis of variance for the CMs by including TNNT2 expression as an additional factor. The results show that the proportion of variance explained by the individual component does not change and that the median percent variance explained by TNNT2 expression is 6% for expression and 3% for splicing. These results are presented in the Results section and in Figure 2—figure supplement 1

(supplemental figure 9 in version with tracked changes).

"To investigate whether this result may reflect variation in the purity of the CMs, we considered the expression of the gene TNNT2 which encodes for the cardiac muscle troponin T. […] The results show that the proportion of variance explained by the individual component does not change and that the median percent variance explained by TNNT2 expression is 6% (Figure 2—figure supplement 1). "